# Is Self-Repair a Silver Bullet for Code Generation?

**Theo X. Olausson**[1,†], **Jeevana Priya Inala**[2], **Chenglong Wang**[2],
**Jianfeng Gao**[2], **Armando Solar-Lezama**[1]
[1]MIT CSAIL    [2]Microsoft Research

## Abstract

Large language models have shown remarkable aptitude in code generation, but still struggle to perform complex tasks. Self-repair—in which the model debugs and repairs its own code—has recently become a popular way to boost performance in these settings. However, despite its increasing popularity, existing studies of self-repair have been limited in scope; in many settings, its efficacy thus remains poorly understood. In this paper, we analyze Code Llama, GPT-3.5 and GPT-4's ability to perform self-repair on problems taken from HumanEval and APPS. We find that when the cost of carrying out repair is taken into account, performance gains are often modest, vary a lot between subsets of the data, and are sometimes not present at all. We hypothesize that this is because self-repair is bottlenecked by the model's ability to provide feedback on its own code; using a stronger model to artificially boost the quality of the feedback, we observe substantially larger performance gains. Similarly, a small-scale study in which we provide GPT-4 with feedback from human participants suggests that even for the strongest models, self-repair still lags far behind what can be achieved with human-level debugging.

## 1 Introduction

Large language models (LLMs) have proven capable of generating code snippets from natural language specifications, but still struggle on complex coding challenges such as those found in competitions and professional software engineering interviews. Recent work has sought to improve performance by leveraging self-repair (Gupta et al., 2020; Le et al., 2022; Chen et al., 2023b; Zhang et al., 2023), in which the model introspects and corrects mistakes in its own code. Figure 1 shows a typical workflow. First, a program is sampled from a code generation model; this program is then run on a suite of unit tests provided as part of the specification; if the program fails any test, then the error message and the faulty program are given to a feedback generation model, which outputs a short explanation of why the code failed; finally, the feedback is passed to a repair model, which generates a fixed version of the program.[1] On the surface, this is a very attractive idea. It allows the system to overcome mistakes caused by unfortunate samples during decoding; easily incorporates feedback during the repair phase from symbolic systems such as compilers, static analysis tools, and execution engines; and mimics the trial-and-error way in which human software engineers write code.

However, it is important to remember that self-repair requires more invocations of the model, thus increasing the computational cost. In particular, whether self-repair is a winning strategy or not ultimately boils down to whether you would—at an equivalent compute budget—have had a greater chance of success if you had simply drawn more code samples i.i.d. from the model and checked them against the suite of unit tests provided as part of the task. Crucially, in a competitive programming setting the efficacy of self-repair depends not only on the model's ability to generate code, which has been studied extensively in the literature, but also on its ability to identify how the code (generated by the model itself) is wrong with respect to the task specification. As far as we are aware, no previous work has studied the effect of this stage in detail.

---

[1]In practice, generating feedback and producing the corrected code can be done through a single interaction with the model; as we will see, it can still be useful to conceptually treat them as separate steps.

[†]Correspondence to theoxo@csail.mit.edu. Work partially done while T.X.O. was at Microsoft Research. Code and data available at github.com/theoxo/self-repair.

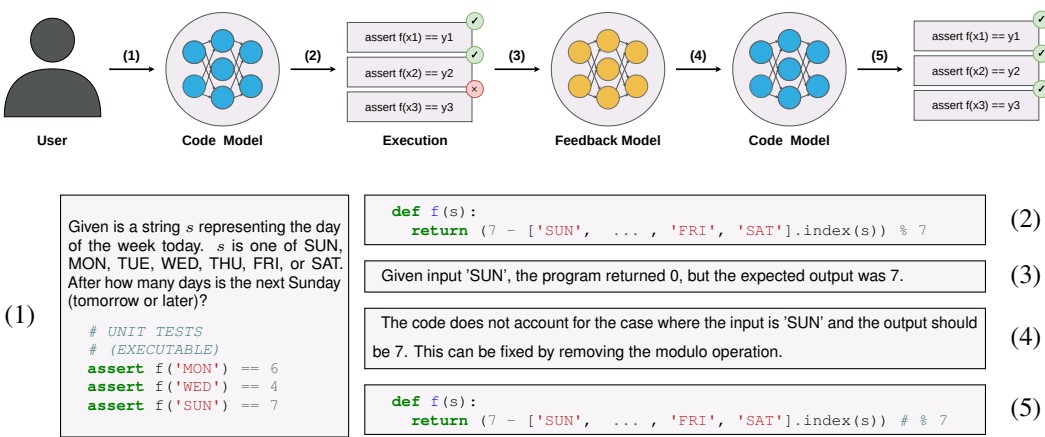

Figure 1: Self-repair with separate code and feedback models. First, a user gives a specification in the form of text and a suite of unit tests (1). Then, a code model (blue) generates a program (2). The program is checked against the unit tests using a symbolic execution engine, and an error message is returned (3). In order to provide more signal to the code model, textual feedback as to *why* this happened is provided by a feedback model (yellow; 4). Finally, this feedback is used by the code model to repair the program (5).

**Contributions:** In this paper, we investigate the efficacy of self-repair techniques applied to CodeLlama-13b-instruct (Rozière et al., 2023), GPT-3.5 (Ouyang et al., 2022; OpenAI, 2022), and GPT-4 (OpenAI, 2023) for self-contained Python programming tasks. We focus on evaluating the models' capacity to reflect upon, provide feedback on and debug the code. We observe that:

- Self-repair is not a silver bullet: when the cost of repair is taken into account, we find several instances in which pass rates are higher or equally high with i.i.d. sampling (without repair), especially when the budget is small. We conjecture that this is because program generation and repair rates trend together, and many subtle factors influence which one will overpower the other for a given task (see Appendix C).

- Self-repair is more likely to be beneficial when more of the sampling budget is spent on generating a diverse set of initial programs than on carrying out extensive repair. For example, for GPT-4 on APPS, drawing 10 samples up front and then 1 repair candidate each (up to 20 samples total) leads to a pass rate $1.05\times$ higher than pass@20 from the same model without repair (from 63.1% to 66.1%); drawing 2 samples up front and then drawing 10 repair candidates each (up to 22 samples total) leads to a pass rate which is *lower* than the baseline pass@22 ($0.97\times$; 63.4% → 61.8%).

- Artificially boosting the quality of the feedback significantly improves the efficacy of self-repair. We replace Code Llama's feedback with that produced by GPT-3.5 or GPT-4, and GPT-3.5's feedback with that of GPT-4; in every case, the boosted configuration beats both the corresponding i.i.d. baseline and the corresponding self-repair configuration at all budgets. Furthermore, replacing GPT-4's own explanations with those of a human programmer improves repair significantly, increasing the fraction of repaired programs which pass the tests by a factor of $1.58\times$ (33.3% → 52.6%).

## 2 RELATED WORK

**Program synthesis with large language models.** The use of large language models for program synthesis has been studied extensively in the literature (Li et al., 2022; Austin et al., 2021; Chen et al., 2021; Le et al., 2022; Fried et al., 2023; Nijkamp et al., 2023; Chowdhery et al., 2022; Touvron et al., 2023; Li et al., 2023). This literature has predominantly focused on evaluating models in terms of either raw accuracy or the pass@k metric (Kulal et al., 2019; Chen et al., 2021), often leveraging filtering techniques based on execution (Li et al., 2022; Shi et al., 2022) or ranking (Chen et al., 2021; Inala et al., 2022; Zhang et al., 2022) to reduce the number of samples which are considered for the final answer. Our work differs from some of the work in this literature in that we assume access to the full collection of input-output examples, as is typically done in inductive synthesis (Kitzelmann, 2010;

Polozov & Gulwani, 2015; Gulwani et al., 2017; Chen et al., 2019a; Ellis et al., 2021). In particular, unlike some prior work (Li et al., 2022; Shi et al., 2022), we do not make a distinction between public tests used for filtering and private tests used to determine correctness, since our method does not involve filtering the outputs.

**Code repair.** Statistical and learning-based code repair has a rich history in both the programming languages and machine learning communities, although it has predominantly been applied to code written by humans in a software engineering context (Long & Rinard, 2016; Bader et al., 2019; Le Goues et al., 2021; Yasunaga & Liang, 2021; Chen et al., 2019b; Mesbah et al., 2019; Wang et al., 2018). More recently, using repair as a post-processing step to improve code which was itself automatically synthesised has been used in the synthesis of both domain-specific languages (Gupta et al., 2020) and general-purpose code (Le et al., 2022; Yasunaga & Liang, 2021; 2020). Our contribution differs from most prior work in this literature in the use of textual feedback for repair, which is possible thanks to the above mentioned rise in the use of LLMs for program synthesis.

**Contemporary work on LLM self-repair.** There is much contemporary work seeking to self-repair with LLMs, both in code generation and beyond. We now highlight a few of these works which are particularly close to ours; see Pan et al. (2023) for a more complete survey of recent work in this quickly evolving field. Zhang et al. (2023) explore self-repair without natural language feedback on APPS (Hendrycks et al., 2021) using both finetuned models and prompt-based self-repair with Codex (Chen et al., 2021), InCoder (Fried et al., 2023), and CodeGen (Nijkamp et al., 2023). Notably, their framework does not consider the cost associated with feedback and repair, which presents a significantly different perspective. Similarly, Chen et al. (2023b) assess Codex's ability to self-repair across a variety of tasks, in a framework that closely resembles that which we study in this work. However, their study differs from ours in terms of the models considered and, more importantly, the research goal, as we specifically aim to investigate the significance of the textual feedback stage. Outside of code generation, self-repair has been used for a wide array of purposes, including mitigating hallucinations and improving factual grounding in search assistants (Peng et al., 2023) as well as code optimization and readability improvements (Madaan et al., 2023). Ultimately, we see our work, in which we investigate the significance of the textual feedback stage in particular, as being complementary to contemporary research which seeks to evaluate self-repair in a broader context; we are eager to see what the implications of our results will be in these other domains.

## 3 METHODOLOGY

### 3.1 SELF-REPAIR OVERVIEW

As shown in Figure 1, we model self-repair as consisting of four stages: code generation, code execution, feedback generation, and code repair. We now formally define these different stages.

**Code generation.** Given a specification $\psi$, a programming model $M_P$ first generates $n_p$ samples i.i.d., which we denote

$$\{p_i\}_{i=1}^{n_p} \overset{i.i.d.}{\sim} M_P(\psi)$$

**Code execution.** These $n_p$ code samples are then executed against a test bed.[2] If any sample $p$ passes all of the tests—which we denote $p \models \psi$—we stop, since a satisfying program has then been found. Otherwise, we collect the error messages $\{e_i\}_i$ returned by the execution environment. These error messages either contain the compile/runtime error information or an example input on which the program's output differs from the expected one. An example is shown in Figure 1 (component 3).

**Feedback generation.** Error messages from the execution environment are usually very high-level, providing little signal for repair. Therefore, as an intermediate step, we use a feedback model to produce a more detailed explanation of what went wrong; Figure 1 (component 4) shows an example. Formally, in this stage, we generate $n_f$ feedback strings, $\{f_{ij}\}_j$, for each wrong program, $p_i$, as follows:

$$\{f_{ij}\}_{j=1}^{n_f} \overset{i.i.d.}{\sim} M_F(\psi; p_i; e_i)$$

Having an explicit feedback generation step allows us to ablate this component so that we can study its significance in isolation.

---

[2] We assume access to the full set of tests in executable form; see Section 5 for a brief discussion on the validity of this assumption in software engineering domains.

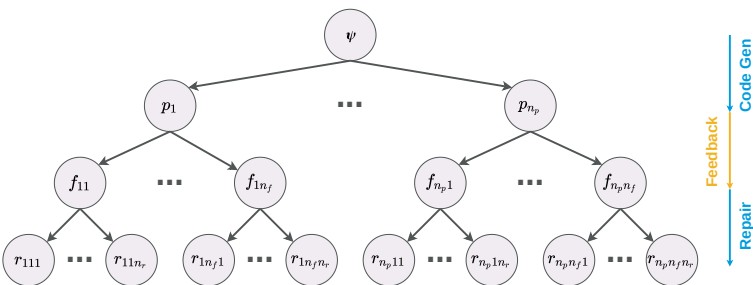

Figure 2: A repair tree begins with a specification $\psi$ (root node), then grows into initial programs $\{p_i\}$, feedback $\{f_{ij}\}$, and repairs $\{r_{ijk}\}$.

**Code repair.** In the final step, for each initial program $p_i$ and feedback $f_{ij}$, $n_r$ candidate repaired programs are sampled from $M_P$ [3]:

$$\{r_{ijk}\}_{k=1}^{n_r} \overset{i.i.d.}{\sim} M_P(\psi; p_i; e_i; f_{ij})$$

**Repair tree.** We call the tree of interleaved text and programs produced by this procedure—rooted in the specification $\psi$, then branching into initial programs $p_i$, each of which branches into feedback $f_{ij}$ and then repairs $r_{ijk}$—a *repair tree*, $T$ (Figure 2).

**Jointly sampling feedback and repair.** The general framework presented above does not require the programming model and feedback model to be the same, thus allowing for the use of specialized models in the system. When $M_P = M_F$, we jointly generate both the feedback and the repaired program in a single sample from the model; see Appendix G for a detailed look at how the prompt differs between this and the previous setting. Formally, we denote this as

$$\{(f_{ij}, r_{ij})\}_{j=1}^{n_{fr}} \overset{i.i.d.}{\sim} M_P(\psi; p_i; e_i)$$

## 3.2 PASS@K FOR SELF-REPAIR

In program synthesis without self-repair, performance is typically measured by pass@k (Chen et al., 2021; Kulal et al., 2019)—the probability that at least one of $k$ i.i.d. program samples from the model satisfies a given specification. In self-repair, program samples are drawn from the model both during the initial sample stage and during the repair stage; thus, we need to adopt pass@k to take into account the number of samples from both stages.

In the main body of this work, we treat repair trees $T$ as themselves forming independent samples from a joint model $T \sim M = (M_P \circ M_F \circ M_P)$ and define the number of programs in the tree as $|\text{programs}(T)| \triangleq n_p + n_p n_{fr}$ (or $|\text{programs}(T)| \triangleq n_p + n_p n_f n_r$); we then compare against a baseline with $k = |\text{programs}(T)|$ i.i.d. samples. We believe this will make our findings most relevant to practitioners, who are likely to deploy self-repairing agents with batched sampling. Appendix A repeats our experiments with two alternative evaluation strategies, in which we vary the search strategy and measure sampling cost by the total number of tokens sampled from the model to take into account the varying lengths of feedback and program samples. Importantly, although the details differ, the overall trends which we observe remain the same.

Independently generating a large amount of repair trees for each setting of the hyper-parameters quickly becomes computationally infeasible, so we plot bootstrapped estimates of the pass rates in our experiments. We first generate a single very large repair tree for each task specification, with: $N_p \geq n_p$ initial program samples; $N_f \geq n_f$ feedback strings per wrong program; and $N_r \geq n_r$ repair candidates per feedback string. Given a setting of $(n_p, n_f, n_r)$, we then sub-sample (with replacement) $N_t = 1000$ different sub-repair-trees from this frozen dataset and average over the runs. We use $N_p = 50$ for all experiments, and consider $n_p \leq 25$ for the self-repair approaches and $n_p \leq 50$ for the baseline, no-repair approach. Similarly, for the feedback strings, we use $N_f = 25$

---

[3]We use the same model for both the initial code generation and the code repair, since these are fundamentally similar tasks.

and $n_f \leq 10$ (except for Section 4.2, in which we only consider $n_f = 1$ and therefore settle for $N_f = 10$ instead). Finally, for the repair candidates we set $N_r = n_r = 1$, since we do joint sampling of feedback and repair in most of our experiments. Estimating the pass rates in this way greatly reduces the computational cost of our experiments, since we can reuse the same initial dataset to compute the estimates for all of the various choices of $n_p, n_f$, and $n_r$.

## 4 EXPERIMENTS

In this section, we carry out experiments to answer the following research questions: (a) In the context of Python programming puzzles, is self-repair better than i.i.d. sampling without repair for the models we consider? If so, under what hyper-parameters is self-repair most effective? (b) Would a stronger feedback model boost the model's repair performance? (c) Would keeping a human in the loop to provide feedback unlock better repair performance even for the strongest model?

We evaluate these hypothesis for two API-served models—GPT-3.5 (Ouyang et al., 2022; OpenAI, 2022) and GPT-4[4] (OpenAI, 2023)—as well as CodeLlama-13b-instruct[5] (Rozière et al., 2023), a model with publicly accessible weights which can be run locally on consumer-level hardware. We consider Python programming challenges from both APPS (Hendrycks et al., 2021) and HumanEval (Chen et al., 2021); on APPS, in order to keep our experiments tractable, we evaluate on a randomly chosen set of 300 tasks.[6] We implement self-repair using templated string concatenation with one-shot prompting; our prompts are given in Appendix G. Based on preliminary experiments, we set the decoding temperature to 0.8 for all models. When appropriate, we compare against a baseline without repair. This baseline, shown with a black line in the plots, is simply i.i.d. sampling from the corresponding model (e.g., GPT-4 when we explore whether GPT-4 is capable of self-repair).

### 4.1 SELF-REPAIR IS NOT A SILVER BULLET, BUT IMPROVES WITH DIVERSE INITIAL SAMPLES

In this subsection, we consider the setup where $M_P = M_F$, i.e., a true self-repair setting in which a single model is used for both code/repair generation and feedback generation. To evaluate if self-repair leads to better performance than a no-repair, i.i.d. sampling-based baseline approach, we vary $n_p$ and $n_{fr}$—that is, the number of initial i.i.d. base samples and joint feedback, repair samples drawn from $M_P$—in the range $(n_p, n_{fr}) \in \{1, 2, 5, 10, 25\} \times \{1, 3, 5, 10\}$.[7]

Figure 4 shows the results for Code LLama and GPT-3.5 on HumanEval, while Figure 3 shows the results for GPT-3.5 and GPT-4 on the more challenging APPS dataset. (We also run GPT-4 on HumanEval and CodeLlama on APPS, but defer these results to Appendix B for brevity.) In the left-hand subplots, the color of each dot indicates the number of initial samples ($n_p$), while its shape indicates the number of feedback-repair samples ($n_{fr}$). In the right hand plots, we show a heat-map with the two hyper-parameters along the axes, where the value in each cell indicates the mean pass rate with self-repair normalized by the mean pass rate of the baseline, no-repair approach when given the same budget. When the normalized mean pass rate is 1, this means that self-repair achieves the same pass rate as the no-repair, baseline approach at that same sample budget; a higher value ($\geq 1$) means self-repair performs better than the baseline.

On APPS, we observe marginal gains for GPT-3.5 only for the largest values of $n_p$. GPT-4, on the other hand, shows more significant improvements, beating out the baseline by up to 8%. When we break the problems down by their difficulty level (see figures in Appendix C), we find that gains are larger on harder problems: GPT-3.5 sees up to a 34% performance gain relative to the baseline on competition-level problems, for example. Meanwhile, on HumanEval we observe performance gains similar to those of GPT-4 on APPS for Code Llama (up to 10% improvement relative to the baseline), while gains for GPT-3.5 are limited as it approaches the ceiling (up to 3%).

---

[4]We use the frozen endpoints `gpt-3.5-turbo-0301` and `gpt-4-0314`.

[5]https://huggingface.co/codellama/CodeLlama-13b-Instruct-hf

[6]These tasks are proportionally sampled in accordance with the frequency of the different difficulty levels in the broader APPS test set: 180 interview-level questions, 60 competition-level questions, and 60 introductory-level questions. All tasks are listed in Appendix H.

[7]Recall that when $M_P = M_F$, we jointly sample for $n_{fr}$ pairs of feedback strings and repair programs instead of sampling them one after another (Section 3.1).

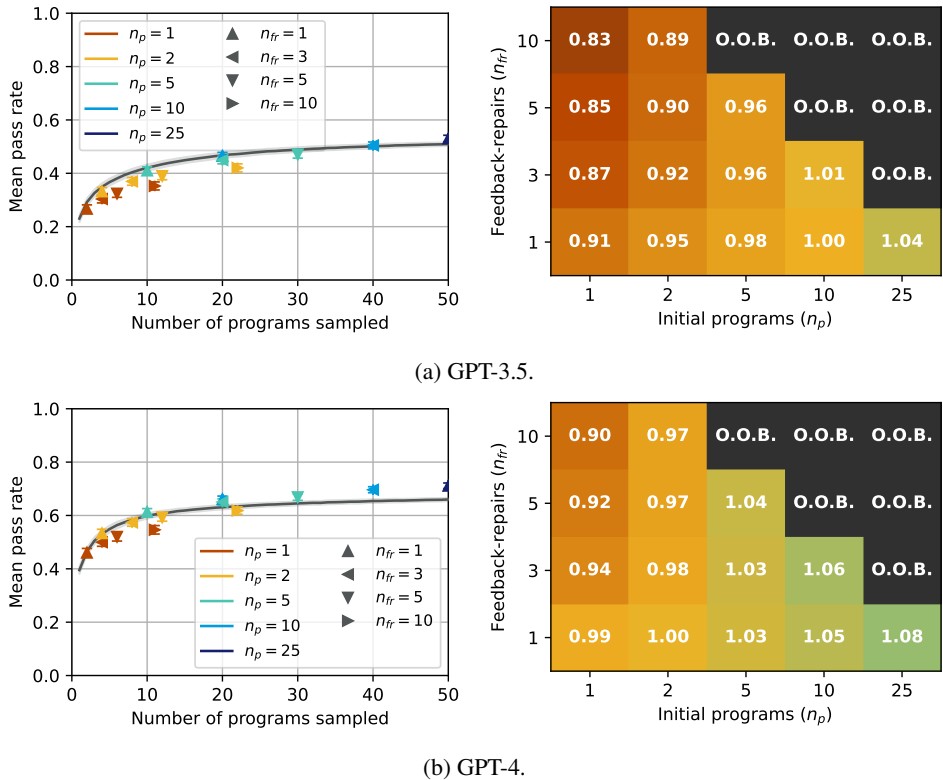

(a) GPT-3.5.

(b) GPT-4.

Figure 3: GPT-3.5 and GPT-4 self-repair results on **APPS**. *Left:* Mean pass rate vs. number of samples generated. Black line is i.i.d. sampling without repair from the same model. Note that the error bars are often smaller than the markers. *Right:* Normalized mean pass rate relative to the baseline at an equivalent budget. Cells for which the number of samples exceeds 50 marked O.O.B. (out of bounds).

From these observations, it is clear that self-repair is not always the best strategy when compared to a non-repair baseline with the same sample budget, especially for smaller budgets. Moreover, it is hard to predict *when* self-repair will be effective. In an analysis of the repair success rates (Appendix C), we find that stronger models have higher repair success rates on easier tasks—but at the same time, the chance of getting a correct program by resampling also increases the easier a task is. Therefore, we see that program generation and repair success rates trend together, and many subtle unknown factors influence which one will overpower the other on any given domain.

While the overall efficacy of self-repair is unclear, we do observe a clear trend with respect to the relationship between the hyper-parameters. Given a fixed number of feedback-repairs ($n_{fr}$), increasing the number of initial programs ($n_p$) (i.e., moving right along the x-axis on the heat maps) consistently leads to relative performance gains for all models. On the other hand, fixing $n_p$ and increasing $n_{fr}$ (i.e., moving up along the y-axis on the heat maps) does not appear to be worth the additional cost incurred, giving marginal gains at higher budgets and oftentimes even decreasing performance at lower budgets. This suggests that, given a fixed budget, the most important factor determining whether self-repair will lead to a correct program or not is the diversity of the base samples that are generated up-front, rather than the diversity of the repairs sampled. Having more initial samples increases the likelihood of there being at least one program which is close to the ideal program and, hence, can be successfully repaired.

Since $n_{fr} = 1$ appears to be the best overall choice for the hyper-parameter $n_{fr}$, we next isolate the effect of the number of initial programs, $n_p$, by exploring a denser set of possible values: $(n_p, n_{fr}) \in \{1, 2, ...., 24, 25\} \times \{1\}$. The plots are shown in Figure 5 for $M_P = M_F \in$

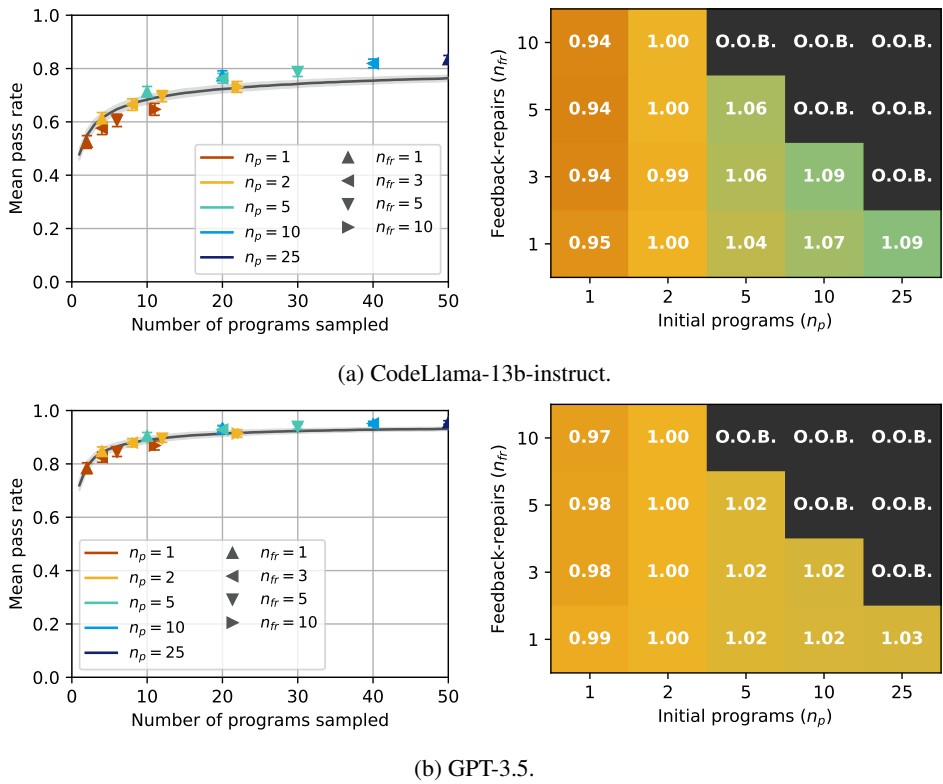

(a) CodeLlama-13b-instruct.

(b) GPT-3.5.

Figure 4: CodeLlama-13b-instruct and GPT-3.5 self-repair results on **HumanEval**. *Left:* Mean pass rate vs. number of samples generated. Black line is i.i.d. sampling without repair from the same model. Note that the error bars are often smaller than the markers. *Right:* Normalized mean pass rate relative to the baseline at an equivalent budget. Cells for which the number of samples exceeds 50 marked O.O.B. (out of bounds).

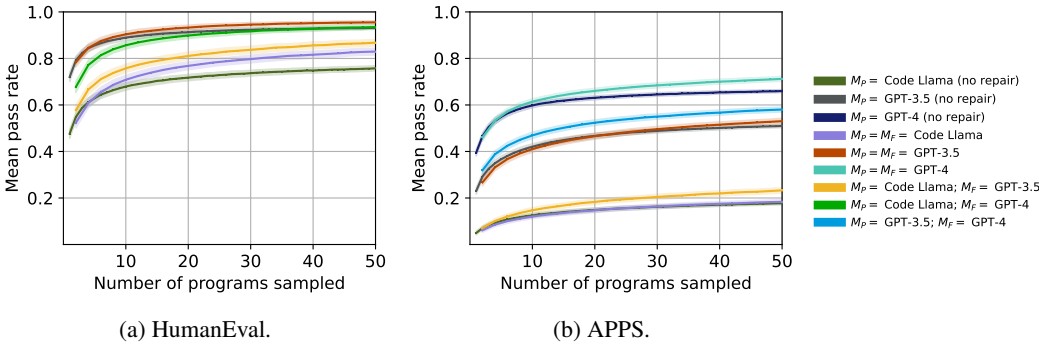

(a) HumanEval.

(b) APPS.

Figure 5: Results when $n_{fr}$ (or $n_f$ and $n_r$) = 1. Shaded region shows $\pm 1$ standard deviation.

{CodeLlama, GPT-3.5, GPT-4} and the baseline, no-repair approaches.[8][9] We observe performance gains for both Code Llama and GPT-3.5 on HumanEval. On APPS, only GPT-4 significantly benefits from self-repair, while both Code Llama and GPT-3.5 mostly lag behind or match their baselines, possibly seeing some very marginal gains at high budgets. In all cases, performance gains at smaller budgets are very marginal or non-existant, but grow somewhat as the budget increases.

---

[8]As GPT-3.5 is already near ceiling on HumanEval, we omit GPT-4 from this figure to reduce clutter.

[9]Note that since $n_{fr}$ is fixed, in these plots, there is a direct correlation between $n_p$ and $k$: $k = n_p + n_p$.

Table 1: Success rate of repair with GPT-4's explanations vs. with those of our human participants.

| Difficulty | Introductory | Interview | Competition | Overall |
|---|---|---|---|---|
| GPT-4 Feedback | 42.64% | 19.33% | 3.67% | 33.30% |
| Human Feedback | 62.21% | 45.67% | 14.67% | 52.60% |

## 4.2 BOOSTING THE FEEDBACK UNLOCKS PERFORMANCE GAINS FROM REPAIR

Next, we conduct an experiment in which we evaluate the impact of using a separate, stronger model to generate the feedback; this is to test the hypothesis that self-repair is held back by the model's inability to introspect and debug its own code. We thus set $M_P$ to be a weaker model (Code Llama on HumanEval, Code Llama or GPT-3.5 on APPS) and $M_F$ to be a stronger model (GPT-3.5 or GPT-4 for Code Llama on HumanEval; GPT-3.5 for Code Llama and GPT-4 for GPT-3.5 on APPS). We then vary the hyper-parameters as $(n_p, n_f, n_r) \in \{1, ...., 25\} \times \{1\} \times \{1\}$, similarly to the previous experiment.[10] [11]

The results for this experiment are also shown in Figure 5 (Code Llama paired with GPT-3.5 in yellow; Code Llama with GPT-4 in bright green; GPT-3.5 with GPT-4 in bright blue). We observe a consistent trend: on APPS, both Code Llama and GPT-3.5 now beat out both their baselines (dark green, gray) and their respective self-repair modes (purple, red). On HumanEval, the performance that Code Llama gains increases further with the strength of the feedback model; note in particular the performance that Code Llama gains when given feedback from GPT-4 (bright green line). This suggests that the textual feedback stage itself is of crucial importance, and that improving it relieves the bottleneck in self-repair.

## 4.3 HUMAN FEEDBACK SIGNIFICANTLY IMPROVES THE SUCCESS RATE OF GPT-4 REPAIR

For our final experiment, we consider the effect of using an expert human programmer's feedback when performing repair with very strong models such as GPT-4. The goal of this study is not to do a direct comparison between a human-in-the-loop approach vs. self-repair, since a human-in-the-loop approach imposes more cognitive burden, which we do not study. Instead, our goal is to further investigate how and why feedback quality affects downstream performance in self-repair.

**Data collection methodology.** We recruit 16 participants and collect a total of 2 human-written pieces of feedback for each of 40 failing programs sampled from GPT-4. Each program is shown to two different participants, to reduce variance caused by participants' skill levels and writing style. Participants were asked to spend approximately one hour on the study overall, and were compensated with a $15 gift card. This study was approved by our Institutional Review Board (IRB) and carried out exclusively through an online survey. See Appendix D for more details on the data collection methodology, including a complete copy of the instructions which we provide to our participants.

**Quantitative analysis.** Having obtained two human-written pieces of feedback for each program, we sample 25 repair candidates for each (feedback, program)-pair from GPT-4. We condition on the specification, the initial program, and the feedback string; in addition to the feedback collected from our participants, we also try two of GPT-4's own feedback strings for each program. Finally, we execute all of these candidate repairs against the test bed, and take note of how often they pass.

The results are summarized in Table 1, with a complete task-by-task breakdown in Appendix E. We note that the overall success rate is increased by over $1.58\times$ when we replace GPT-4's own feedback with that of our human participants. Perhaps unsurprisingly, the relative difference increases as the problems get harder, indicating that GPT-4's ability to produce accurate and useful feedback trails further behind our human participants' when the task (and code) becomes more complex.

**Qualitative analysis.** We manually go through all of GPT-4's and the participants' feedback and note down whether the feedback: (a) seems, at a cursory glance, to be correct, or if it is obviously inaccurate; (b) explicitly suggests a small change to the code (e.g. "change the condition on line

---

[10]Note that since we are now operating in a setting in which the feedback and repair stages must be separated, we have three hyper-parameters—$n_p, n_f, n_r$—instead of two—$n_p, n_{fr}$ (Section 3.1).

[11]To reduce cost, we use $N_f = 10$ instead of $N_f = 25$ for this experiment (see Section 3.2).

X"); (c) explicitly suggests a large change to the code (e.g. "frame the problem as min-cut instead of shortest-path"); (d) contains blocks of pseudocode or Python (which GPT-4's feedback never does, per our experiment design); or (e) expresses uncertainty (using phrases such as "unsure", "it appears", etc.).[12] Examples of each category are shown in Appendix F. We find that

- Almost all human-contributed feedback interleaves natural language with occasional single-statement math/code expressions; only 2/80 responses include pseudocode or explicit Python.
- GPT-4's feedback is more likely to be obviously inaccurate (32/80 vs. 7/80 for the human feedback).
- GPT-4 is more likely to explicitly suggest small changes (54/80 vs. 42/80 for GPT-4 and the participants, respectively; 28/48 vs. 38/73 if we filter out suggestions which are obviously incorrect), while human participants show a slightly greater tendency to suggest high-level changes (23/80 vs. 18/80 for GPT-4; 21/73 vs. 13/48 when seemingly correct).
- Our human participants sometimes express uncertainty (7/80); GPT-4 never does (0/80).

This further analysis suggests that the results in Table 1 are not due to artefacts such as our participants providing explicit code blocks which the model simply copies. Instead, the difference in performance appears to be caused by a combination of more accurate feedback, a greater ability to suggest high-level, large-scale changes to the code when needed, and our participants' ability to express their uncertainty (instead of confidently giving potentially inaccurate feedback).

## 5 LIMITATIONS

Firstly, to reduce computational cost, we pre-populate and then sub-sample from a single large repair tree to bootstrap a large number of repair trees for each setting of the hyper-parameters (Section 3.2). This risks introducing statistical artefacts in our analysis. To minimize this risk, we bounded $n_p$ and $n_{fr}$ far below $N_p$ and $N_{fr}$, respectively, in our self-repair experiments. Furthermore, we note that the standard deviation is very small in our experiments for all values of $n_p$ and $n_{fr}$ (see the scatter plots in Figures 3, 4), offering increased confidence in our results.

Secondly, our experiments focus on self-contained Python programming tasks with executable unit tests. This is quite different from real-world software development tasks, where specifications are often incomplete, there are long contextual dependencies, and tests are unlikely to be available for each individual snippet. Future work will be required to see what role self-repair can play there: for example, whether it could resolve ambiguities in the specification, or if automatic unit test synthesis techniques (Li et al., 2022; Chen et al., 2023a) could be leveraged alongside established engineering practices like Test-Driven Development (Astels, 2003) to overcome the lack of high quality tests.

Finally, our study on human data did not track how much time the participants took to debug the programs. As a result, we can only evaluate the quality of the feedback (and the impact this has on repair). Further research at the intersection of Human-Computer Interaction, AI, and program synthesis is needed to explore when and how human intervention should be leveraged, as well as how programming assistants should be designed to facilitate this style of interaction.

## 6 CONCLUSION

We investigated self-repair for code generation, looking specifically at CodeLlama-13b-instruct, GPT-3.5 and GPT-4 on problems taken from HumanEval and APPS. In a series of experiments, we observed that (1) when the cost of carrying out repair is taken into account, performance gains from self-repair are often modest, vary not only between but also within datasets, and rely on achieving sufficient diversity in the initial programs. Furthermore, by replacing the feedback stage we found that (2) substituting a weaker model's own feedback with that of a stronger model significantly improved performance. Finally, we carried out an experiment with human participants, in which we found that (3) replacing GPT-4's self-generated feedback with feedback provided by an experienced programmer increased the number of repaired programs which pass all unit tests by $1.58\times$. Our results suggest that self-repair is not a silver bullet for code generation, and that current models are held back by their inability to reliably produce accurate and useful feedback on why the code is wrong.

---

[12] We do not count individual single-line statements/expressions such as "$x = 5$" as pseudocode or Python.

ACKNOWLEDGEMENTS

T.X. Olausson is supported by the Defense Advanced Research Projects Agency (DARPA) under the ASKEM program, award HR00112220042. T.X. Olausson was also supported through a position at Microsoft Research for part of the time period during which this work was carried out. A. Solar-Lezama is supported by the National Science Foundation (NSF) and Intel Corporation through NSF Grant CCF:2217064. This work benefited greatly from discussion with several colleagues at Microsoft Research. Any opinions, findings, and conclusions or recommendations expressed in this material are those of the author(s) and do not necessarily reflect the views of the National Science Foundation, the Defense Advanced Research Projects Agency, Intel Corporation, or Microsoft Research.

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

**Data:** Task $\psi$; sample budgets $n_p, n_f, n_r$
**Result:** A tuple (success, token count)
$P \leftarrow [M_P(\psi) \mid i \leftarrow 0 \text{ to } n_p]$;
$t \leftarrow \text{sum}([\text{num\_tokens}(p) \mid p \in P])$;
**if** $any([p \models \psi \mid p \in P])$ **then**
    |   **return** *(True, t)*;
**end**
$R \leftarrow []$;
**for** $p \in P$ **do**
    |   $e \leftarrow \text{error\_msg}(p, \psi)$;
    |   $F_p \leftarrow [M_F(\psi; p; e) \mid i \leftarrow 0 \text{ to } n_f]$;
    |   $t \leftarrow t + \text{sum}([\text{num\_tokens}(f) \mid f \in F_p])$;
    |   **for** $f \in F$ **do**
    |   |   $R_{pf} \leftarrow [M_P(\psi; p; e; f) \mid i \leftarrow 0 \text{ to } n_r]$;
    |   |   $t \leftarrow t + \text{sum}([\text{num\_tokens}(r) \mid r \in R_{pf}])$;
    |   |   $R \leftarrow R + R_{pf}$
    |   **end**
**end**
**if** $any([r \models \psi \mid r \in R])$ **then**
    |   **return** *(True, t)*;
**end**
**return** *(False, t)*;

**Algorithm 1:** Generating a repair tree $T$, computing $T \models \psi$ and its token count with **batched** self-repair. All operations should be taken to run in parallel whenever possible.

**Data:** Task $\psi$; sample budgets $n_p, n_f, n_r$
**Result:** A tuple (success, token count)
$t \leftarrow 0$;
**for** $i \leftarrow 1 \text{ to } n_p$ **do**
    |   $p_i \leftarrow M_P(\psi)$;
    |   $t \leftarrow t + \text{size}(p_i)$;
    |   **if** $p_i \models \psi$ **then**
    |   |   **return** *(True, t)*;
    |   **end**
    |   $e_i \leftarrow \text{error\_msg}(p_i, \psi)$;
    |   **for** $j \leftarrow 1 \text{ to } n_f$ **do**
    |   |   $f_{ij} \leftarrow M_F(\psi; p_i; e_i)$;
    |   |   $t \leftarrow t + \text{size}(f_{ij})$;
    |   |   **for** $k \leftarrow 1 \text{ to } n_r$ **do**
    |   |   |   $r_{ijk} \leftarrow M_P(\psi; p_i; e_i; f_{ij})$;
    |   |   |   $t \leftarrow t + \text{size}(r_{ijk})$;
    |   |   |   **if** $r_{ijk} \models \psi$ **then**
    |   |   |   |   **return** *(True, t)*;
    |   |   |   **end**
    |   |   **end**
    |   **end**
**end**
**return** *(False, t)*;

**Algorithm 2:** Generating a repair tree $T$, computing $T \models \psi$ and its token count with **sequential** self-repair. All operations executed serially.

## A  ALTERNATIVE EVALUATION STRATEGIES FOR SELF-REPAIR

In the main part of this paper, we chose to evaluate self-repair in terms of an adapted version of `pass@k` (Chen et al., 2021; Kulal et al., 2019), in which a single repair tree is considered equivalent to $k = n_p + n_p * n_{fr}$ samples from the baseline. This makes the results easy to digest for practitioners and scholars who are familiar with `pass@k` and prior work in this literature. However, this evaluation strategy does not account for the feedback tokens produced by the same model, which also come at a cost, and so risks overemphasizing the benefits of self-repair.

In this appendix, we present and briefly discuss results in terms of two alternative evaluation strategies which address the non-uniform costs of program and feedback samples by comparing two dependent variables—the pass rate and the number of tokens which had to be sampled from the model in order to achieve it—an approach which we dub **pass@t**. This allows us to compare not only how successful a particular configuration is but also how much "work" it requires from the model.

Formally, suppose that you are given a dataset $D = \{\psi_d\}_d$ and a chosen set of values for the hyper-parameters $(M_P, M_F, n_p, n_f, n_r)$. Let $T_d^i \sim M(\psi_d)$ denote a repair tree that is sampled as described in Section 3.1 for the task $\psi_d$; let `num_tokens`$(T_d^i)$ denote the total number of program and feedback tokens in the repair tree; and say that $T_d^i \models \psi_d$ is true if, and only if, $T_d^i$ has at least one leaf program that satisfies the unit tests in the specification $\psi_d$. Then the `pass@t` metric of this choice of hyper-parameters is defined as the expected pass rate at the number of tokens which you would expect to generate with this choice of hyper-parameters:

$$\texttt{pass@t} \triangleq \mathop{\mathbb{E}}_{\substack{\psi_d \sim D \\ T_d^i \sim M(\psi_d)}} \left[ T_d^i \models \psi_d \right] \quad \textbf{at} \quad t = \mathop{\mathbb{E}}_{\substack{\psi_d \sim D \\ T_d^i \sim M(\psi_d)}} \left[ \texttt{num\_tokens}(T_d^i) \right]$$

## A.1 BATCHED PASS@T

The first variation we will consider is *batched* $pass@t$. In this strategy, repair trees are assumed to be generated as in Algorithm 1: all $n_p$ initial programs are sampled in parallel, then checked for correctness; if none of them pass, then all $n_p * n_{fr}$ repairs of all initial programs are sampled in parallel, after which we check if any of the repairs pass. The total number of tokens sampled so far is recorded at every point, and returned alongside the value of $T \models \psi$. Thus, the number of tokens which are sampled depends on both the success rate in the initial round of program generation as well as the relative verbosity of the feedback and programs. Averaging the results over all of the tasks, we get not only a mean pass rate but also a mean token count, which can be plotted together as points on a curve.

Figures 6, 7 and 8 show the results of all experiments from main paper, repeated with this evaluation strategy. Note that while these plots may at first look much like those of Section 4 they are subtly different in that *both* axes are now dependent variables (recall that in $pass@k$, $k$ is an independent variable set ahead of time). The better a particular model is, the closer it would thus get to $(0.0, 1.0)$—i.e. the top-left corner of the plot.

Broadly speaking, we observe the same trends as were noted in Section 4: gains for GPT-4 on APPS as well as both Code Llama and GPT-3.5 on HumanEval; larger gains when the feedback is provided by the stronger model; typically better performance when setting $n_p > n_{fr}$, except for GPT-3.5 on HumanEval where performance is relatively stable across the board as it is already near ceiling.

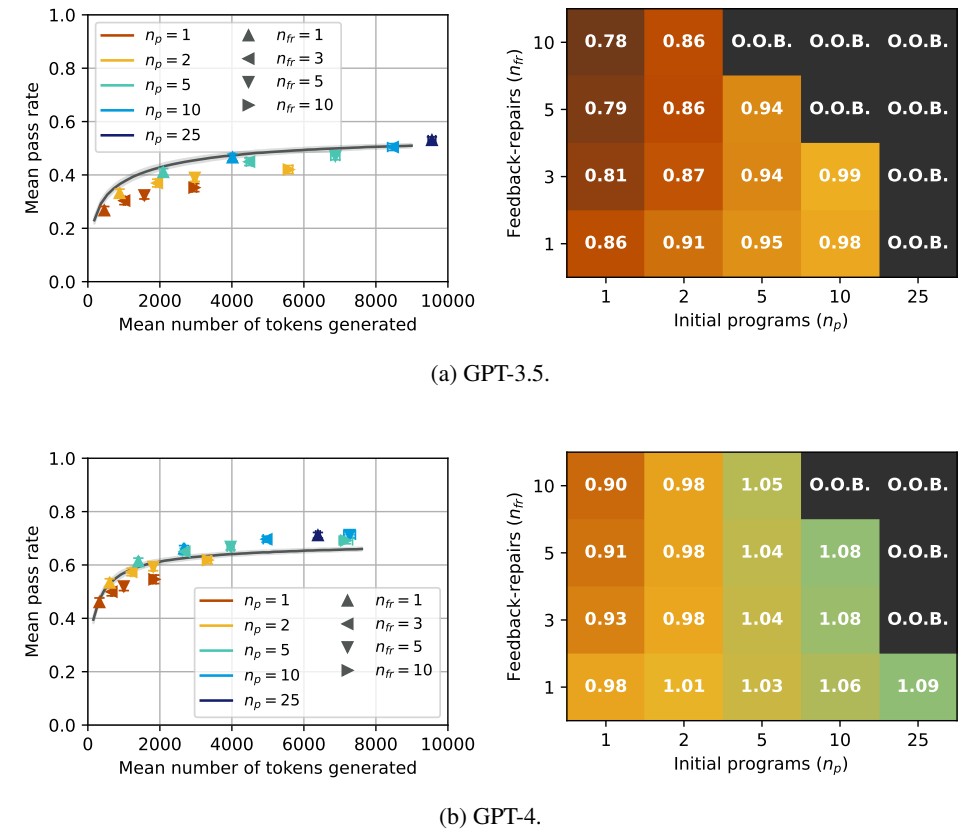

(a) GPT-3.5.

(b) GPT-4.

Figure 6: GPT-3.5 and GPT-4 self-repair results on APPS, evaluated in terms of **batched pass@t**. C.f. Figure 3. N.B.: The heatmaps here display the normalized mean pass rate relative to the (interpolated) baseline at an equivalent number of tokens.

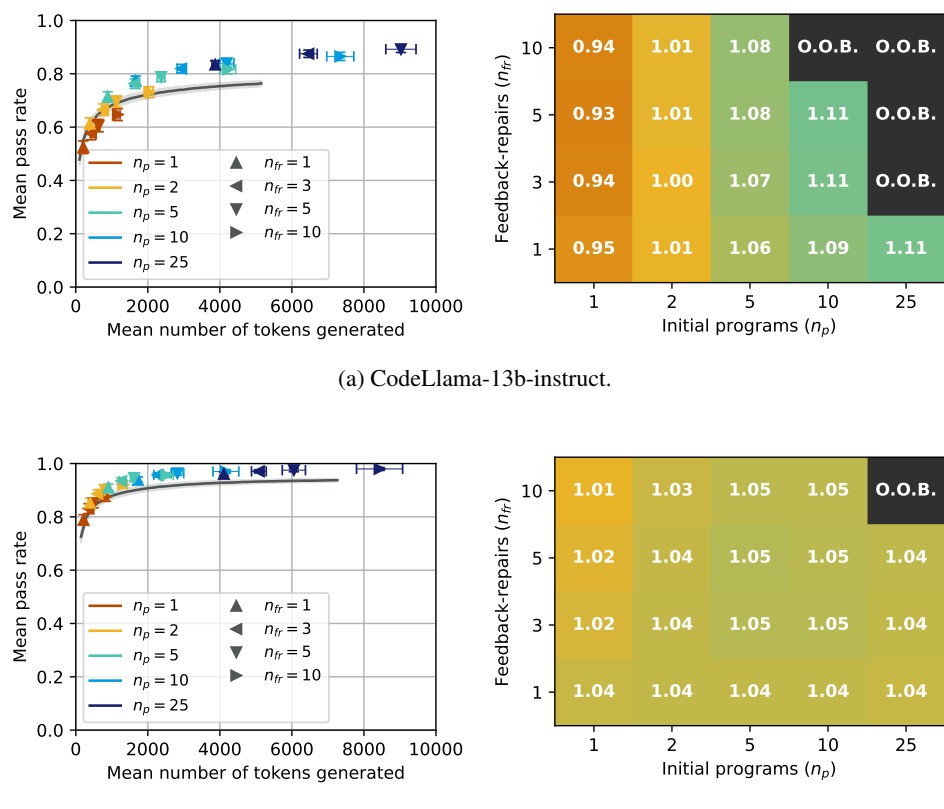

(a) CodeLlama-13b-instruct.

(b) GPT-3.5.

Figure 7: CodeLlama-13b-instruct and GPT-3.5 self-repair results on HumanEval, evaluated in terms of **batched `pass@t`**. C.f. Figure 4. N.B.: The heatmaps here display the normalized mean pass rate relative to the (interpolated) baseline at an equivalent number of tokens.

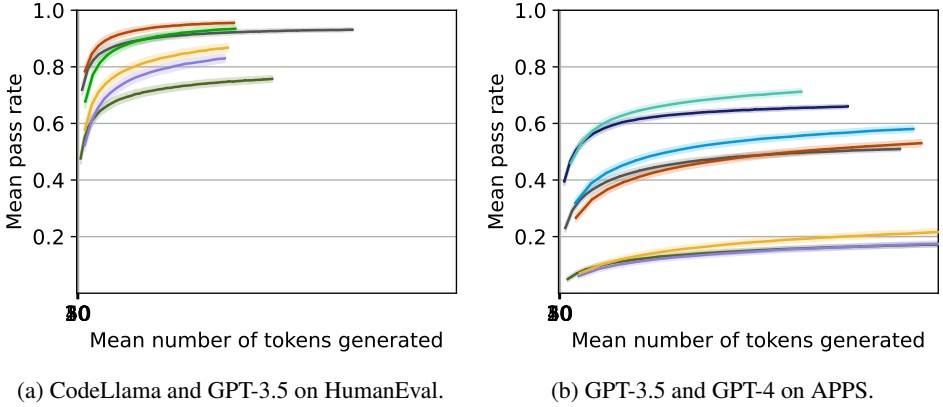

(a) CodeLlama and GPT-3.5 on HumanEval.

(b) GPT-3.5 and GPT-4 on APPS.

Figure 8: **Batched `pass@t`** curves for each model when $n_{fr}$ (or $n_f$ and $n_r$) = 1. C.f. Figure 5.

## A.2 SEQUENTIAL PASS@T

The batched sampling approaches considered in Section 4 and A.1 are designed to mimic the way in which practitioners are likely to deploy large-scale self-repair without user intervention. However, this is quite different from the way in which end-users interact with chat-based programming assistants. This is likely to take on a more sequential form, where the user first receives a single program, spends some time trying to get the assistant to debug it, before finally giving up and starting over from scratch in a new session. One might be curious how our results extend to such a setting.

In this section, we model self-repair as a depth-first search for a passing program, where the parameters $n_p, n_f, n_r$ are taken to be *bounds* on the widths of each level; this is shown in Algorithm 2. Note that this even more tightly couples the observed pass rates and the number of tokens generated: if the pass rate is high, a passing program will quickly be found and the number tokens generated will be low, and vice versa.

We again repeat the experiments from the main paper: the results are shown in Figures 9, 10, 11. As before, the key trends are still discernible. However, in this setting, self-repair appears to be somewhat less beneficial; especially when the baseline pass rate is already high. This is particularly visible when comparing the heatmaps in Figures 9 and 10 to those from before (e.g., 6, 7), as well as Figure 11.

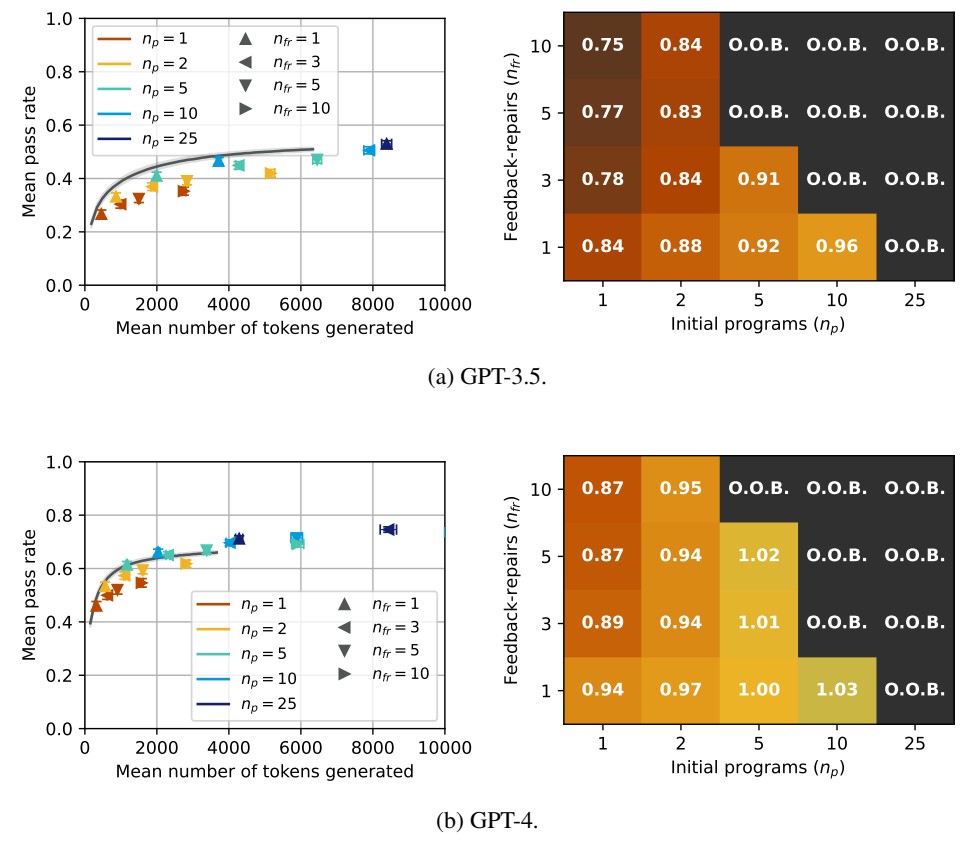

(a) GPT-3.5.

(b) GPT-4.

Figure 9: GPT-3.5 and GPT-4 self-repair results on APPS, evaluated in terms of **sequential pass@t**. C.f. Figure 3. N.B.: The heatmaps here display the normalized mean pass rate relative to the (interpolated) baseline at an equivalent number of tokens.

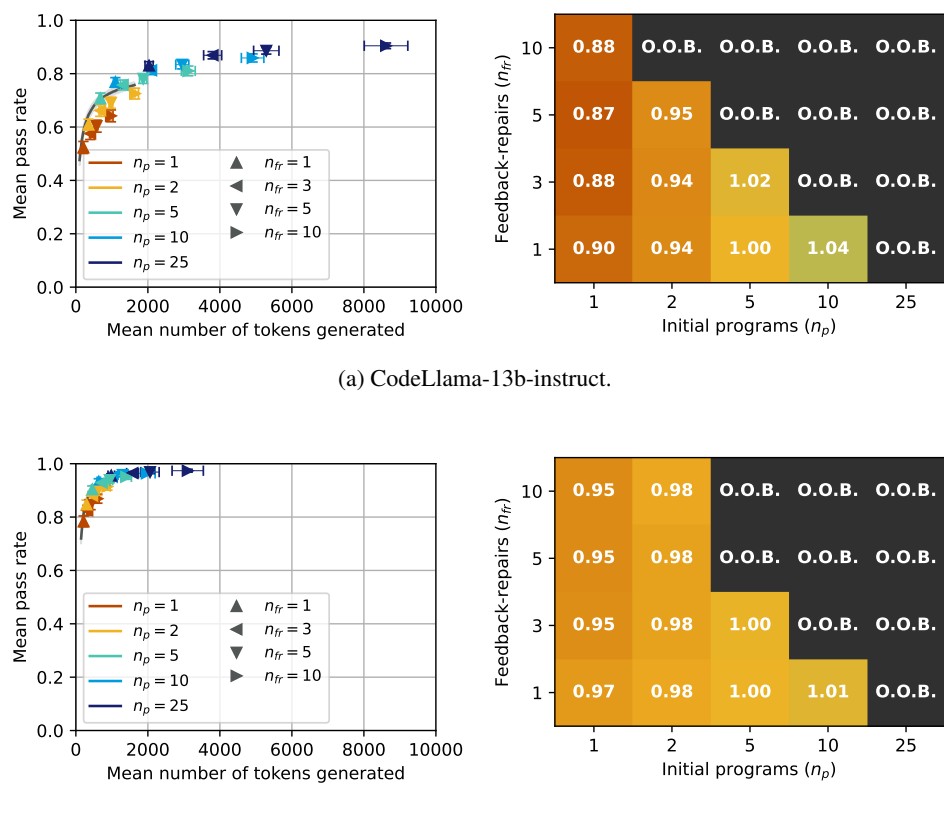

Figure 10: CodeLlama-13b-instruct and GPT-3.5 self-repair results on HumanEval, evaluated in terms of **sequential** `pass@t`. C.f. Figure 4. N.B.: The heatmaps here display the normalized mean pass rate relative to the (interpolated) baseline at an equivalent number of tokens.

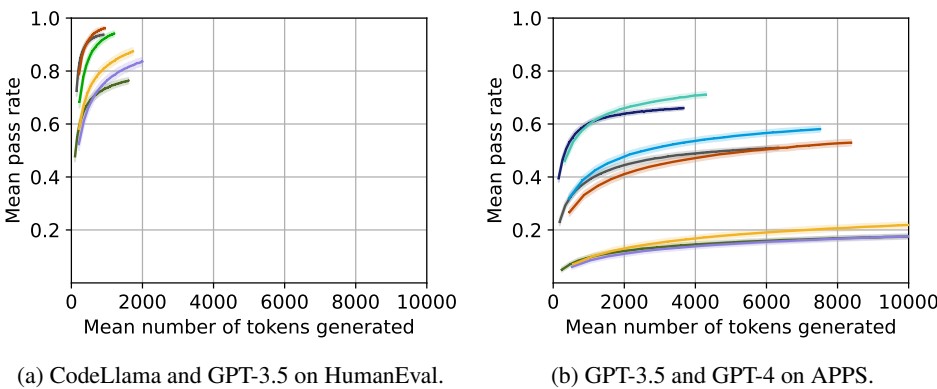

(a) CodeLlama and GPT-3.5 on HumanEval.      (b) GPT-3.5 and GPT-4 on APPS.

Figure 11: **Sequential** `pass@t` curves for each model when $n_{fr}$ (or $n_f$ and $n_r$) = 1. C.f. Figure 5.

# B ADDITIONAL RESULTS: GPT-4 ON HUMANEVAL, CODE LLAMA ON APPS

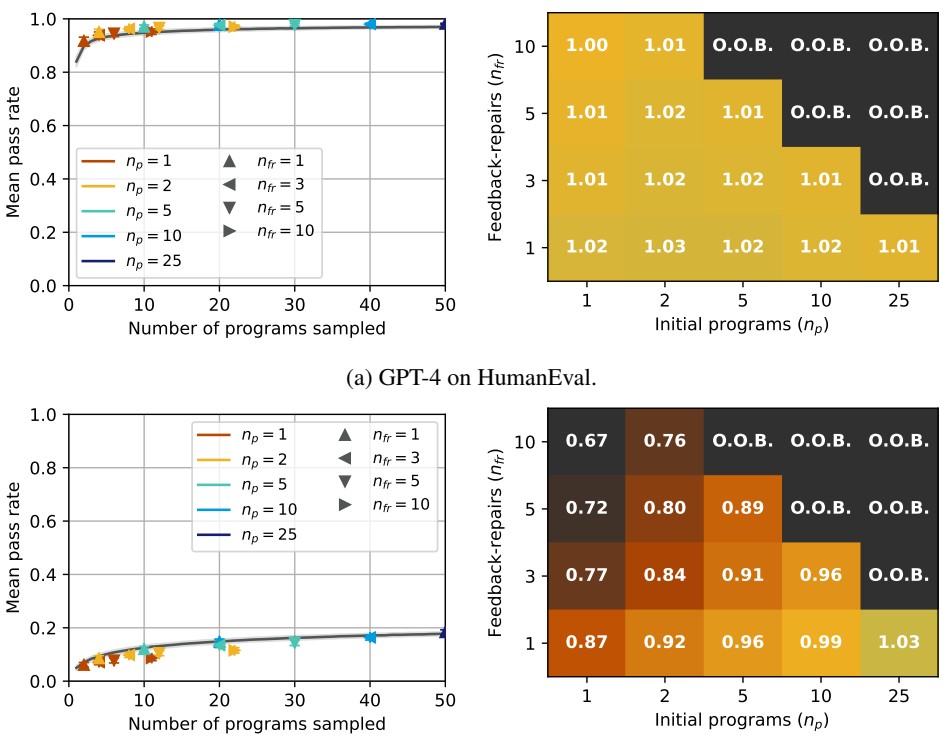

(a) GPT-4 on HumanEval.

(b) CodeLlama-13b-instruct on APPS.

Figure 12: Full Code Llama APPS and GPT-4 HumanEval results. Omitted from Section 4.1 for brevity.

# C   ADDITIONAL RESULTS: SELF-REPAIR VS. PROBLEM DIFFICULTY

APPS problems are divided into three categories: `introductory`, `interview` and `competition`. This makes it easy to repeat our APPS experiments on problems of a specific difficulty; the results are shown in Figures 13 through 16. We note that both GPT-3.5 and GPT-4 appear to benefit more from self-repair the harder the problem is. Meanwhile, Code Llama benefits *less*; we also note that GPT-3.5's baseline performance on APPS-introductory problems (Figure 14, top) is very similar to that of GPT-3.5 on HumanEval (Figure 4b), yet self-repair only appears fruitful in the latter experiment. The relationship between baseline performance and the efficacy of self-repair thus appears to not be so clear cut.

One might also want to evaluate the success rate of repair without the confounding factor of how often the initial sample of programs passes the tests; intuitively, we expect that harder programs should be harder to repair. Table 2 shows the fraction of repaired programs which pass the tests on APPS. Although it is important not to place too much weight on the specific numbers, since—for example—a less performant model's initial programs might be more difficult to repair than those generated by a stronger model, these results agree with our intuition.

We leave it to future work to investigate in detail why self-repair performance gains do not appear to trend perfectly with baseline performance; we offer the conjecture that it is due to a combination of (a) the power struggle between feedback generation and repair success rate (which benefit self-repair) vs. program generation success rate (which benefits i.i.d. sampling without repair); (b) the prevalence of ambiguity in the natural language specification, which might affect self-repair's ability to correctly identify flaws in a failing program; and (c) the informativeness of the unit tests. In the meantime, as has been shown in this work, improving the model's ability to provide feedback on code (e.g. through finetuning on code explanation data) can boost the performance of self-repair.

Table 2: Repair success rates in various settings. The repair success rate is computed as `number_of_passing_repairs` / `total_number_of_repairs_sampled`.

| Dataset | Difficulty | Model | Repair Success Rate |
|---------|------------|-------|---------------------|
| APPS | introductory | CL-13b-instruct | 2.8% |
| | | CL-13b-instruct+GPT-3.5 | 5.4% |
| | | GPT-3.5 | 13.7% |
| | | GPT-3.5+GPT-4 | 29.1% |
| | | GPT-4 | 28.8% |
| | interview | CL-13b-instruct | 1.0% |
| | | CL-13b-instruct+GPT-3.5 | 1.9% |
| | | GPT-3.5 | 4.2% |
| | | GPT-3.5+GPT-4 | 11.2% |
| | | GPT-4 | 8.7% |
| | competition | CL-13b-instruct | 0.1% |
| | | CL-13b-instruct+GPT-3.5 | 0.4% |
| | | GPT-3.5 | 1.5% |
| | | GPT-3.5+GPT-4 | 3.3% |
| | | GPT-4 | 8.6% |
| | overall | CL-13b-instruct | 1.1% |
| | | CL-13b-instruct+GPT-3.5 | 2.2% |
| | | GPT-3.5 | 4.7% |
| | | GPT-3.5+GPT-4 | 11.5% |
| | | GPT-4 | 10.8% |
| HumanEval | - | CL-13b-instruct | 9.1% |
| | | CL-13b-instruct+GPT-3.5 | 20.1% |
| | | CL-13b-instruct+GPT-4 | 39.3% |
| | | GPT-3.5 | 22.4% |
| | | GPT-4 | 49.6% |

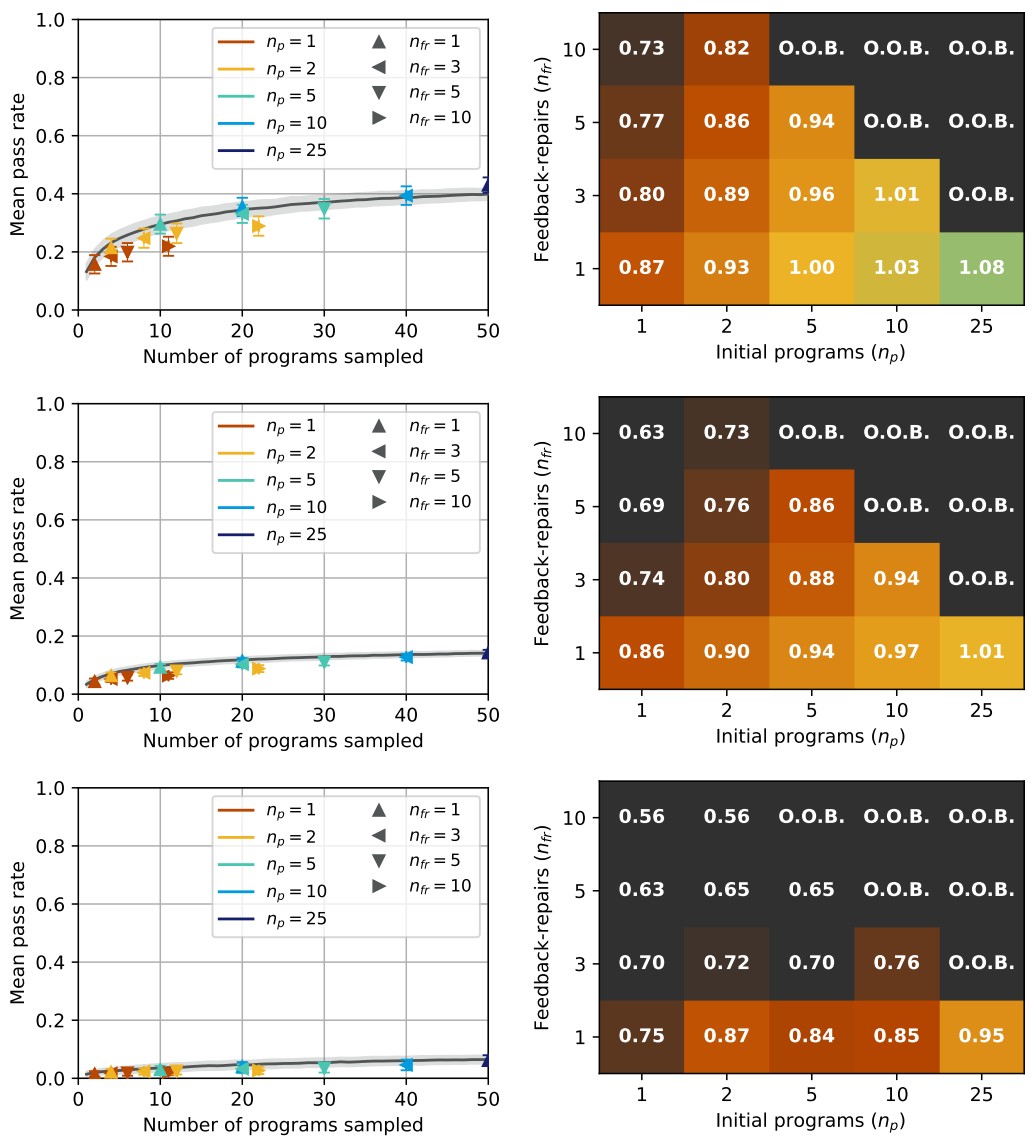

Figure 13: CodeLlama-13b-instruct results from Figure 12b (Appendix B) per APPS difficulty (row), from top to bottom: `introductory`, `interview`, and `competition`.

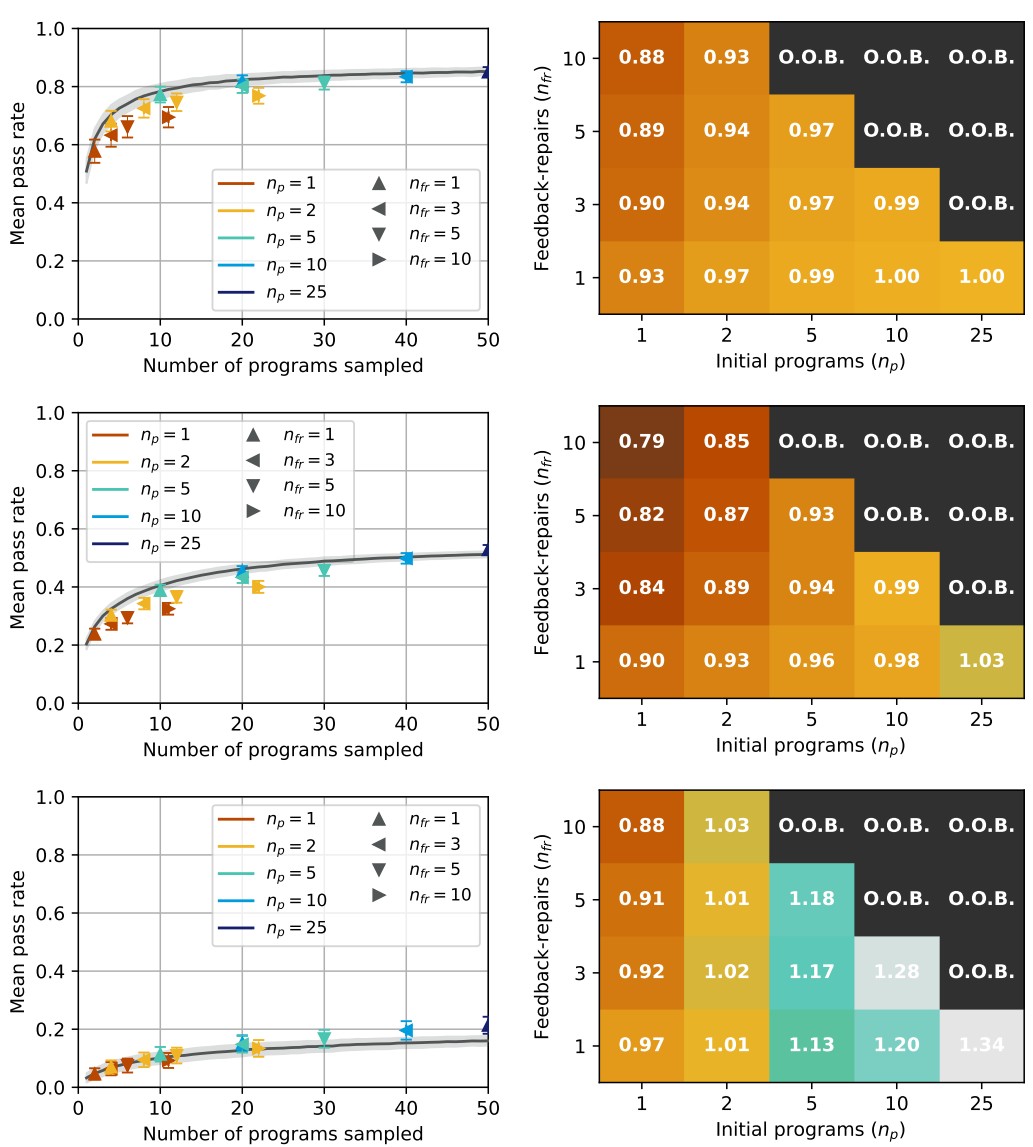

Figure 14: GPT-3.5 results from Figure 3 (Section 4.1) per APPS difficulty (row), from top to bottom: `introductory`, `interview`, and `competition`.

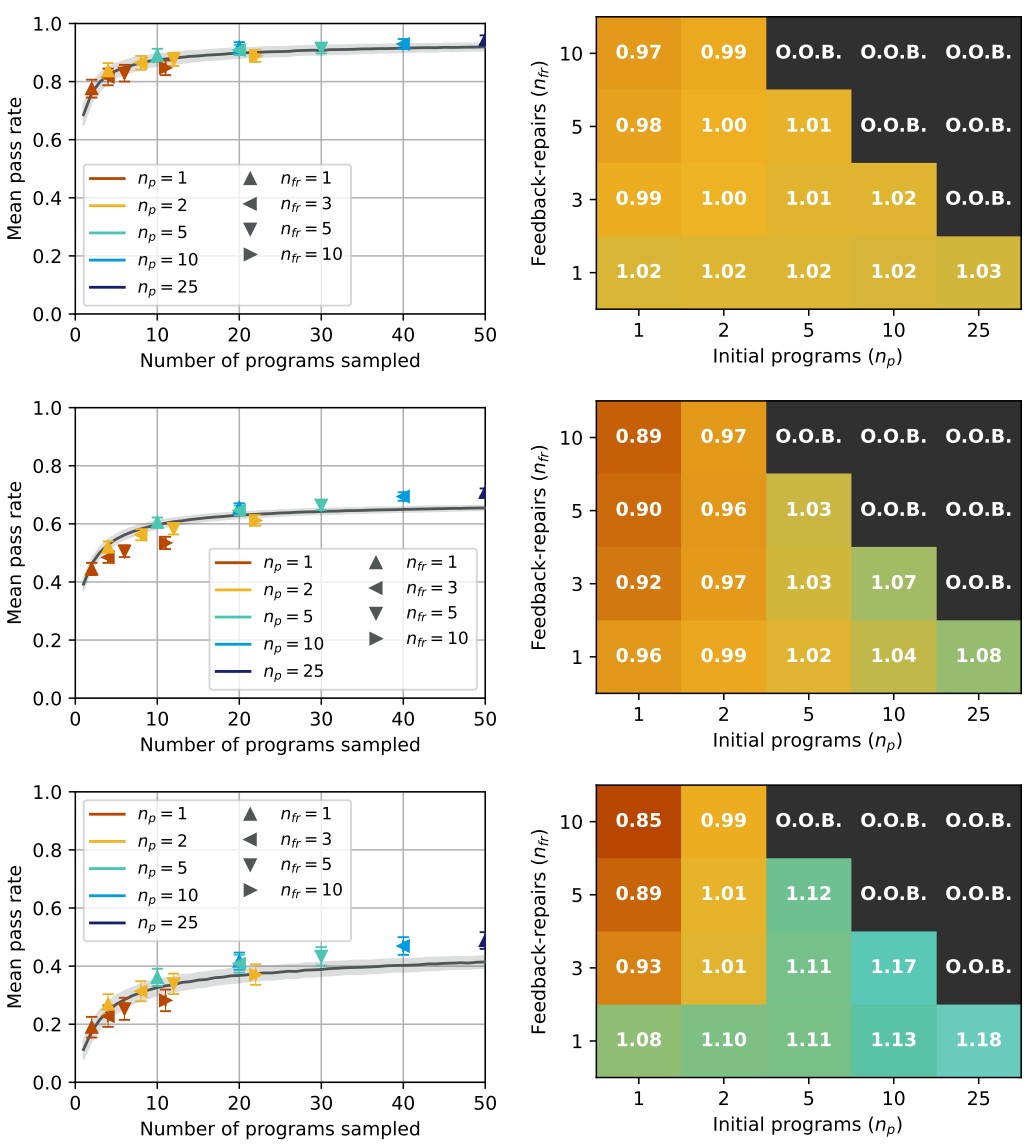

Figure 15: GPT-4 results from Figure 3 (Section 4.1) per APPS difficulty (row), from top to bottom: `introductory`, `interview`, and `competition`.

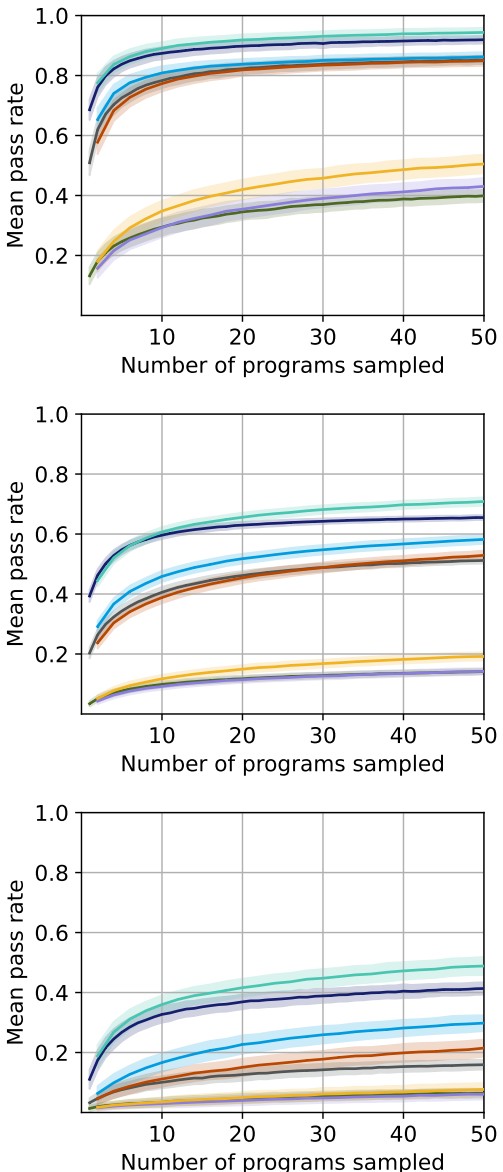

Figure 16: Results from Figure 5b (Section 4.2) per APPS difficulty (row), from top to bottom: `introductory`, `interview`, and `competition`.

## D    HUMAN EXPERIMENT: DETAILS AND STUDY INSTRUCTIONS

**Participants.** We recruit 16 participants, consisting of 15 graduate students and 1 professional machine learning engineer. Participants were told to spend approximately one hour on the study overall, and were compensated with a $15 gift card.

**Data collection.** We first sample 20 tasks $\{\psi_i\}_{i=1}^{20}$ from the APPS test set; to make the data collection process less time-consuming for the participants of the study, we skew the distribution towards easier tasks (14 introductory; 3 interview; 3 competition). For each task $\psi_i$, we then sample two failing GPT-4 completions $p_{i,1}, p_{i,2}$, making for a total of $20 \cdot 2 = 40$ programs to refine. Each participant is provided with five different base programs based on their level of experience with Python and competitive programming. Programs are taken from distinct tasks; participants are never showed two different programs belonging to the same task. Participants are then asked to explain, in their own words, what the program is doing wrong. To reduce the cognitive load for participants, each program $p_{i,j}$ is accompanied by the error message $e_{i,j}$ and two feedback strings $f_{i,j,1}, f_{i,j,2}$ sampled from GPT-4. We obtain these feedback strings by randomly sampling from the feedback-repair pairs used in the previous experiments and removing the code block. Note that each of the 40 programs will be shown to two different participants, to reduce variance caused by participants' skill levels and writing style. This human data collection was approved by our Institutional Review Board (IRB) and carried out exclusively through an online survey.

**Instructions.** Participants were given a slide deck with instructions. The following ten images show the instructions, which include an example of a task shown to a participant:

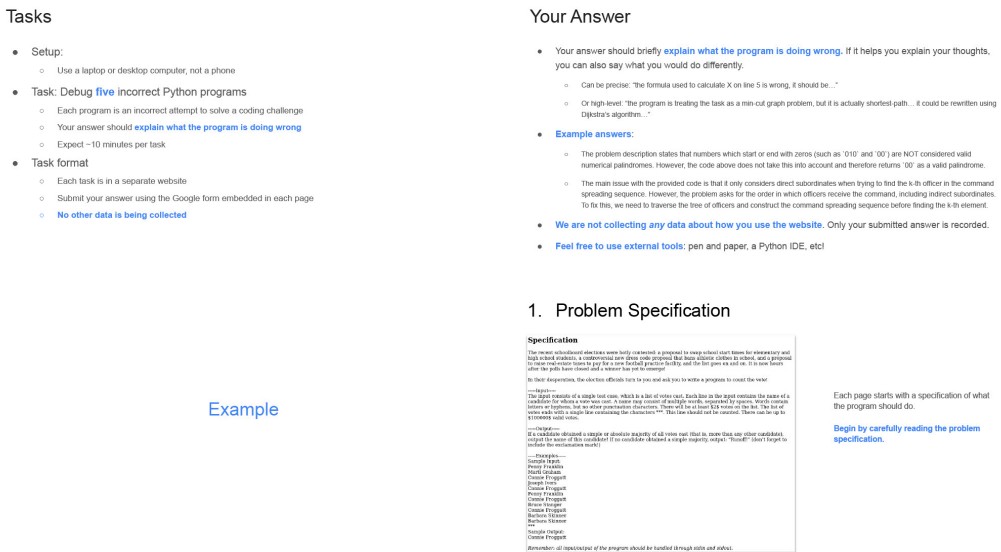

### 2. Incorrect Program

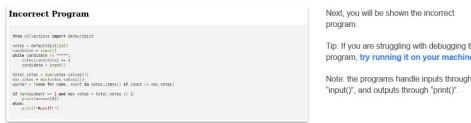

Next, you will be shown the incorrect program.

Tip: If you are struggling with debugging the program, **try running it on your machine!**

Note: the programs handle inputs through "input()", and outputs through "print()".

### 3. Error Message

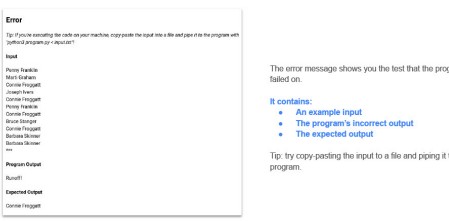

The error message shows you the test that the program failed on.

**It contains:**
- **An example input**
- **The program's incorrect output**
- **The expected output**

Tip: try copy-pasting the input to a file and piping it to the program.

### 4. Model Explanations

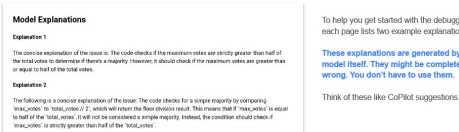

To help you get started with the debugging, each page lists two example explanations.

**These explanations are generated by the model itself. They might be completely wrong. You don't have to use them.**

Think of these like CoPilot suggestions.

### 5. Answer Form

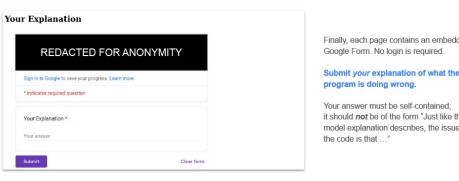

Finally, each page contains an embedded Google Form. No login is required.

**Submit *your* explanation of what the program is doing wrong.**

Your answer must be self-contained; it should **not** be of the form "Just like the first model explanation describes, the issue with the code is that …"

### Study Tips

We are *very* grateful for your help! 😊

- **Make sure you understand the task first!** The programs have subtle logic errors, not just simple compiler errors.
- Try to write **clear and concise** explanations, with proper grammar and punctuation.
- Feel free to **use (or not use) the model explanations** when writing your answers; but make sure your answer is self-contained!
- The tasks vary in difficulty. Feel free to **allocate your time as you see fit**; we are not measuring how quickly you complete the tasks or anything like that!
- Feel free to use **external tools**:
  - Use pen and paper or a whiteboard to help you reason about the task at hand.
  - Use a Python IDE to execute and debug the code.
  - Search online for help.
- **Have a question? Ask ███ before moving on with the study!** 😊

### FAQ

- Are you collecting data as I visit the website?
  - **No** - none at all. Only your final answers are recorded.
- What is the point of the study?
  - To investigate how much better the models are at **fixing code when given human feedback**, instead of having to debug the code themselves.
- Are you evaluating how useful the model explanations were to me?
  - **No** - they are just there to help you get started with the debugging. We only care about your final answer.

# E  HUMAN EXPERIMENT (QUANTITATIVE ANALYSIS): RESULTS PER TASK

In the table below, we give a complete breakdown of the quantitative results presented in Section 4.3.
Note that each program is associated with four different pieces of feedback: two sampled from GPT-4,
and two given by our human participants. Each cell is the number of repair candidates (out of 25)
that passed all the unit tests. See Section 4.3 for details, as well as Appendix D for the instructions
given to participants.

| Task | Difficulty | Program | GPT-4 #1 | GPT-4 #2 | Human #1 | Human #2 |
|------|-----------|---------|----------|----------|----------|----------|
| 2106 | interview | A | 7 | 10 | 10 | 0 |
|      |           | B | 0 | 2 | 20 | 16 |
| 2673 | interview | A | 4 | 7 | 17 | 24 |
|      |           | B | 3 | 25 | 25 | 25 |
| 2923 | interview | A | 0 | 0 | 0 | 0 |
|      |           | B | 0 | 0 | 0 | 0 |
| 3070 | competition | A | 0 | 0 | 0 | 0 |
|      |           | B | 3 | 0 | 5 | 0 |
| 3286 | competition | A | 2 | 6 | 10 | 25 |
|      |           | B | 0 | 0 | 0 | 4 |
| 3754 | competition | A | 0 | 0 | 0 | 0 |
|      |           | B | 0 | 0 | 0 | 0 |
| 4182 | introductory | A | 25 | 25 | 25 | 24 |
|      |           | B | 25 | 0 | 25 | 25 |
| 4195 | introductory | A | 25 | 3 | 24 | 23 |
|      |           | B | 23 | 25 | 25 | 25 |
| 4281 | introductory | A | 0 | 4 | 0 | 0 |
|      |           | B | 0 | 0 | 0 | 0 |
| 4333 | introductory | A | 25 | 0 | 25 | 0 |
|      |           | B | 23 | 24 | 24 | 25 |
| 4347 | introductory | A | 0 | 0 | 7 | 25 |
|      |           | B | 0 | 0 | 25 | 25 |
| 4426 | introductory | A | 25 | 25 | 25 | 25 |
|      |           | B | 25 | 25 | 25 | 25 |
| 4450 | introductory | A | 0 | 0 | 0 | 0 |
|      |           | B | 24 | 0 | 22 | 24 |
| 4507 | introductory | A | 0 | 0 | 0 | 0 |
|      |           | B | 0 | 0 | 1 | 0 |
| 4514 | introductory | A | 15 | 21 | 1 | 16 |
|      |           | B | 0 | 0 | 25 | 0 |
| 4704 | introductory | A | 0 | 25 | 0 | 25 |
|      |           | B | 25 | 25 | 24 | 23 |
| 4741 | introductory | A | 25 | 25 | 25 | 25 |
|      |           | B | 25 | 25 | 25 | 25 |
| 4855 | introductory | A | 0 | 1 | 17 | 25 |
|      |           | B | 0 | 2 | 3 | 23 |
| 4873 | introductory | A | 0 | 0 | 0 | 0 |
|      |           | B | 0 | 0 | 0 | 18 |
| 4952 | introductory | A | 0 | 0 | 2 | 25 |
|      |           | B | 24 | 8 | 24 | 21 |

## F HUMAN EXPERIMENT (QUALITATIVE ANALYSIS): EXAMPLES

In this appendix, we give examples of each category used to classify the responses in the qualitative analysis of Section 4.3. For each category, we give one example from the human participants and one from GPT-4 (when possible). Each example contains: the textual specification of the task; the incorrect program; the error message (in the format it was shown to participants); the feedback in question; and a short comment motivating its classification.

### F.1 CATEGORY (A): OBVIOUSLY INCORRECT FEEDBACK

### F.1.1 GPT-4: EXAMPLE

```
### SPECIFICATION
The recent schoolboard elections were hotly contested: a proposal to swap school start times for elementary
      and high school students, a controversial new dress code proposal that bans athletic clothes in school,
      and a proposal to raise real-estate taxes to pay for a new football practice facility, and the list
      goes on and on. It is now hours after the polls have closed and a winner has yet to emerge!

In their desperation, the election officials turn to you and ask you to write a program to count the vote!

-----Input-----
The input consists of a single test case, which is a list of votes cast. Each line in the input contains the
      name of a candidate for whom a vote was cast. A name may consist of multiple words, separated by
      spaces. Words contain letters or hyphens, but no other punctuation characters. There will be at least
      votes on the list. The list of votes ends with a single line containing the characters ***. This line
      should not be counted. There can be up to valid votes.

-----Output-----
If a candidate obtained a simple or absolute majority of all votes cast (that is, more than any other
      candidate), output the name of this candidate! If no candidate obtained a simple majority, output:
      "Runoff!" (don't forget to include the exclamation mark!)

-----Examples-----
Sample Input:
Penny Franklin
Marti Graham
Connie Froggatt
Joseph Ivers
Connie Froggatt
Penny Franklin
Connie Froggatt
Bruce Stanger
Connie Froggatt
Barbara Skinner
Barbara Skinner
***
Sample Output:
Connie Froggatt

### INCORRECT PROGRAM
from collections import defaultdict

votes = defaultdict(int)
candidate = input()
while candidate != "***":
    votes[candidate] += 1
    candidate = input()

total_votes = sum(votes.values())
max_votes = max(votes.values())
winner = [name for name, count in votes.items() if count == max_votes]

if len(winner) == 1 and max_votes > total_votes // 2:
    print(winner[0])
else:
    print("Runoff!")

### ERROR MESSAGE
===Input===
Penny Franklin
Marti Graham
Connie Froggatt
Joseph Ivers
Connie Froggatt
Penny Franklin
Connie Froggatt
Bruce Stanger
Connie Froggatt
Barbara Skinner
Barbara Skinner
***
===Program Output===
Runoff!
===Expected Output===
Connie Froggatt
```

```
### FEEDBACK
The concise explanation of the issue is: The code checks if the maximum votes are strictly greater than half
    of the total votes to determine if there's a majority. However, it should check if the maximum votes
    are greater than or equal to half of the total votes.

### COMMENT
The issue with the code is not that the ">" should have been a ">=", but rather that it is not allowing for a
    simple majority.
```

## F.1.2   HUMAN: EXAMPLE

```
### SPECIFICATION
Snuke and Raccoon have a heap of N cards. The i-th card from the top has the integer a_i written on it.
They will share these cards.
First, Snuke will take some number of cards from the top of the heap, then Raccoon will take all the
    remaining cards.
Here, both Snuke and Raccoon have to take at least one card.
Let the sum of the integers on Snuke's cards and Raccoon's cards be x and y, respectively.
They would like to minimize |x-y|.
Find the minimum possible value of |x-y|.

-----Constraints-----
- 2 \leq N \leq 2 \times 10^5
- -10^{9} \leq a_i \leq 10^{9}
- a_i is an integer.

-----Input-----
Input is given from Standard Input in the following format:
N
a_1 a_2 ... a_{N}

-----Output-----
Print the answer.

-----Sample Input-----
6
1 2 3 4 5 6

-----Sample Output-----
1

If Snuke takes four cards from the top, and Raccoon takes the remaining two cards, x=10, y=11, and thus
    |x-y|=1. This is the minimum possible value.

### INCORRECT PROGRAM
def main():
    n = int(input())
    a = list(map(int, input().split()))

    Snuke_sum = 0
    Raccoon_sum = sum(a)

    min_diff = abs(Raccoon_sum)
    for i in range(n-1):
        Snuke_sum += a[i]
        Raccoon_sum -= a[i]
        diff = abs(Snuke_sum - Raccoon_sum)
        min_diff = min(min_diff, diff)

    print(min_diff)

main()

### ERROR MESSAGE
===Input===
2
10 -10
===Program Output===
0
===Expected Output===
20

### FEEDBACK
The code initializes the result `min_diff` to `abs(Raccoon_sum)` corresponding to the case where Raccoon
    takes all the cards, however this is incorrect because of the constraint "Snuke and Raccoon have to
    take at least one card". Instead just setting `min_diff` to negative infinity initially would fix it.

### COMMENT
This human participant was on the right track, but note that setting `min_diff` to *negative* infinity would
    cause it to never be updated inside of the for loop, and so the program would in fact always return
    negative infinity.
```

## F.2   CATEGORY (B): FEEDBACK EXPLICITLY SUGGESTS A MINOR CHANGE TO THE CODE

## F.2.1   GPT-4: EXAMPLE

```
### SPECIFICATION
There is a square in the xy-plane. The coordinates of its four vertices are (x_1,y_1),(x_2,y_2),(x_3,y_3) and
    (x_4,y_4) in counter-clockwise order.
(Assume that the positive x-axis points right, and the positive y-axis points up.)
Takahashi remembers (x_1,y_1) and (x_2,y_2), but he has forgot (x_3,y_3) and (x_4,y_4).
Given x_1,x_2,y_1,y_2, restore x_3,y_3,x_4,y_4. It can be shown that x_3,y_3,x_4 and y_4 uniquely exist and
    have integer values.

-----Constraints-----
- |x_1|,|y_1|,|x_2|,|y_2| \leq 100
- (x_1,y_1) != (x_2,y_2)
- All values in input are integers.

-----Input-----
Input is given from Standard Input in the following format:
x_1 y_1 x_2 y_2

-----Output-----
Print x_3,y_3,x_4 and y_4 as integers, in this order.

-----Sample Input-----
0 0 0 1

-----Sample Output-----
-1 1 -1 0

(0,0),(0,1),(-1,1),(-1,0) is the four vertices of a square in counter-clockwise order.
Note that (x_3,y_3)=(1,1),(x_4,y_4)=(1,0) is not accepted, as the vertices are in clockwise order.

### INCORRECT PROGRAM
x1, y1, x2, y2 = map(int, input().split())

x3 = x2 + (y2 - y1)
y3 = y2 - (x2 - x1)
x4 = x1 + (y2 - y1)
y4 = y1 - (x2 - x1)

print(x3, y3, x4, y4)

### ERROR MESSAGE
===Input===
0 0 0 1
===Program Output===
1 1 1 0
===Expected Output===
-1 1 -1 0

### FEEDBACK
The issue with the code is that it doesn't consider the counter-clockwise order of the vertices. To fix this,
    we need to swap the signs of (y2 - y1) and (x2 - x1) in the calculations.

### COMMENT
Swapping the listed signs is a concrete suggestion of a way to fix the code. Since the overall structure of
    the code would remain the same, we count this as a minor change.
```

## F.2.2   HUMAN: EXAMPLE

```
### SPECIFICATION
Given is a string S representing the day of the week today.
S is SUN, MON, TUE, WED, THU, FRI, or SAT, for Sunday, Monday, Tuesday, Wednesday, Thursday, Friday, and
    Saturday, respectively.
After how many days is the next Sunday (tomorrow or later)?

-----Constraints-----
- S is SUN, MON, TUE, WED, THU, FRI, or SAT.

-----Input-----
Input is given from Standard Input in the following format:
S

-----Output-----
Print the number of days before the next Sunday.

-----Sample Input-----
SAT

-----Sample Output-----
1

It is Saturday today, and tomorrow will be Sunday.

### INCORRECT PROGRAM
S = input()
days = ['SUN', 'MON', 'TUE', 'WED', 'THU', 'FRI', 'SAT']
print((7 - days.index(S)) % 7)

### ERROR MESSAGE
===Input===
SUN
```

```
===Program Output===
0
===Expected Output===
7

### FEEDBACK
The logic is incorrect. There should be no modulo operator.

### COMMENT
Suggesting that the modulo operator should be removed is a relatively small, but explicit, change.
```

## F.3   CATEGORY (C): FEEDBACK EXPLICITLY SUGGESTS A LARGE CHANGE TO THE CODE

### F.3.1   GPT-4: EXAMPLE

```
### SPECIFICATION
Takahashi is about to assemble a character figure, consisting of N parts called Part 1, Part 2, ..., Part N
     and N-1 connecting components. Parts are distinguishable, but connecting components are not.
Part i has d_i holes, called Hole 1, Hole 2, ..., Hole d_i, into which a connecting component can be
     inserted. These holes in the parts are distinguishable.
Each connecting component will be inserted into two holes in different parts, connecting these two parts. It
     is impossible to insert multiple connecting components into a hole.
The character figure is said to be complete when it has the following properties:
- All of the N-1 components are used to connect parts.
- Consider a graph with N vertices corresponding to the parts and N-1 undirected edges corresponding to the
     pairs of vertices connected by a connecting component. Then, this graph is connected.
Two ways A and B to make the figure complete are considered the same when the following is satisfied: for
     every pair of holes, A uses a connecting component to connect these holes if and only if B uses one to
     connect them.
Find the number of ways to make the figure complete. Since the answer can be enormous, find the count modulo
     998244353.

-----Constraints-----
- All values in input are integers.
- 2 \leq N \leq 2 \times 10^5
- 1 \leq d_i < 998244353

-----Input-----
Input is given from Standard Input in the following format:
N
d_1 d_2 \cdots d_N

-----Output-----
Print the answer.

-----Sample Input-----
3
1 1 3

-----Sample Output-----
6

One way to make the figure complete is to connect Hole 1 in Part 1 and Hole 3 in Part 3 and then connect Hole
     1 in Part 2 and Hole 1 in Part 3.

### INCORRECT PROGRAM
def main():
    N = int(input())
    d = list(map(int, input().split()))

    MOD = 998244353

    fact = [1] * (N+1)
    for i in range(1, N+1):
        fact[i] = fact[i-1] * i % MOD

    ans = 0
    for ki in range(N):
        ans += d[ki] * fact[ki] % MOD * fact[N-ki-1] % MOD
        ans %= MOD

    ans = ans * fact[N-1] % MOD

    print(ans)

main()

### ERROR MESSAGE
===Input===
3
1 1 3
===Program Output===
18
===Expected Output===
6

### FEEDBACK
```

```
The following is a concise explanation of the issue: The code above calculates the total number of ways to
    connect the parts, but it does not ensure that each connecting component is used only once. This
    results in overcounting the number of ways to make the figure complete. To fix the issue, we need to
    ensure that each connecting component is used exactly once.

### COMMENT
The feedback suggests changing the code so that each connecting component is used exactly once, which does
    not seem to be taken into account in the current approach.
Changing the code to satisfy this part of the specification seems to require a major rewrite.
```

## F.3.2 HUMAN: EXAMPLE

```
### SPECIFICATION
Oh, New Year. The time to gather all your friends and reflect on the heartwarming events of the past year...

$n$ friends live in a city which can be represented as a number line. The $i$-th friend lives in a house with
    an integer coordinate $x_i$. The $i$-th friend can come celebrate the New Year to the house with
    coordinate $x_i-1$, $x_i+1$ or stay at $x_i$. Each friend is allowed to move no more than once.

For all friends $1 \le x_i \le n$ holds, however, they can come to houses with coordinates $0$ and $n+1$ (if
    their houses are at $1$ or $n$, respectively).

For example, let the initial positions be $x = [1, 2, 4, 4]$. The final ones then can be $[1, 3, 3, 4]$, $[0,
    2, 3, 3]$, $[2, 2, 5, 5]$, $[2, 1, 3, 5]$ and so on. The number of occupied houses is the number of
    distinct positions among the final ones.

So all friends choose the moves they want to perform. After that the number of occupied houses is calculated.
    What is the minimum and the maximum number of occupied houses can there be?

-----Input-----

The first line contains a single integer $n$ ($1 \le n \le 2 \cdot 10^5$) - the number of friends.

The second line contains $n$ integers $x_1, x_2, \dots, x_n$ ($1 \le x_i \le n$) - the coordinates of the
    houses of the friends.

-----Output-----

Print two integers - the minimum and the maximum possible number of occupied houses after all moves are
    performed.
-----Examples-----
Input
4
1 2 4 4
Output
2 4
Input
9
1 1 8 8 8 4 4 4 4

Output
3 8

Input
7
4 3 7 1 4 3 3

Output
3 6

-----Note-----

In the first example friends can go to $[2, 2, 3, 3]$. So friend $1$ goes to $x_1+1$, friend $2$ stays at his
    house $x_2$, friend $3$ goes to $x_3-1$ and friend $4$ goes to $x_4-1$. $[1, 1, 3, 3]$, $[2, 2, 3, 3]$
    or $[2, 2, 4, 4]$ are also all valid options to obtain $2$ occupied houses.

For the maximum number of occupied houses friends can go to $[1, 2, 3, 4]$ or to $[0, 2, 4, 5]$, for example.

### INCORRECT PROGRAM
from collections import Counter

def main():
    n = int(input())
    x = list(map(int, input().split()))

    cnt = Counter(x)
    max_houses = 0
    for pos in range(1, n + 1):
        if cnt[pos]:
            max_houses += 1

    occupied = [0] * (n + 2)
    for pos in x:
        if not occupied[pos - 1]:
```

```
            occupied[pos - 1] = 1
        elif not occupied[pos]:
            occupied[pos] = 1
        elif not occupied[pos + 1]:
            occupied[pos + 1] = 1

    min_houses = sum(occupied)

    print(min_houses, max_houses)

main()

### ERROR MESSAGE
===Input===
4
1 2 4 4
===Program Output===
4 3
===Expected Output===
2 4

### FEEDBACK
The code calculates both the minimum and maximum count of houses incorrectly. The correct solution requires
    conducting a tree search to determine the true maximum and minimum. This can be further optimized by
    first sorting the list of occupied houses and using a sliding window to process independent segments of
    the number line.

### COMMENT
The feedback suggests conducting a tree search to determine the max and min, as well as a way to optimize
    this search.
These constitute major changes to the program as it is currently written.
```

## F.4 CATEGORY (D): FEEDBACK CONTAINS BLOCKS OF PSEUDOCODE OR PYTHON CODE

### F.4.1 GPT-4: EXAMPLE

*GPT-4's feedback never contains blocks of Python per our experiment design, in which the code block
is stripped (Section 4.3). We find that GPT-4 never utilizes blocks of pseudocode, either.*

### F.4.2 HUMAN: EXAMPLE

```
### SPECIFICATION
Given is a string S representing the day of the week today.
S is SUN, MON, TUE, WED, THU, FRI, or SAT, for Sunday, Monday, Tuesday, Wednesday, Thursday, Friday, and
    Saturday, respectively.
After how many days is the next Sunday (tomorrow or later)?

-----Constraints-----
- S is SUN, MON, TUE, WED, THU, FRI, or SAT.

-----Input-----
Input is given from Standard Input in the following format:
S

-----Output-----
Print the number of days before the next Sunday.

-----Sample Input-----
SAT

-----Sample Output-----
1

It is Saturday today, and tomorrow will be Sunday.

### INCORRECT PROGRAM
day = input()
days = ['SUN', 'MON', 'TUE', 'WED', 'THU', 'FRI', 'SAT']
index = days.index(day)
result = (7 - index) % 7
print(result)

### ERROR MESSAGE
===Input===
SUN
===Program Output===
0
===Expected Output===
7

### FEEDBACK
before returning the result, we should just check to see if result ==0.
so we just need to add:
if result == 0:
    result = 7
```

```
### COMMENT
The feedback contains a block of Python code that the model can simply copy in order to fix the program.
```

## F.5 CATEGORY (E): FEEDBACK WHICH EXPRESSES UNCERTAINTY

### F.5.1 GPT-4: EXAMPLE

*In the $20 \cdot 2 \cdot 2 = 80$ examples we study, we do not find that GPT-4 ever expresses uncertainty.*

### F.5.2 HUMAN: EXAMPLE

```
### SPECIFICATION
Two kingdoms had been at war for a long time, until the emperor intervened to bring an end to the conflict.
    The territory in question comprises an $M$ by $N$ rectangular grid. At the emperor's insistence, the
    two kings have withdrawn their troops until no two opposing troops are in adjacent squares of the map
    (adjacent being horizontal or vertical - diagonal is not considered).

The emperor proposes to designate certain squares of the map as neutral territory. Neither king will be
    allowed to move troops into those squares, and the emperor's own forces will patrol them to be sure
    that both kings observe these rules.

The emperor is frugal and does not want to commit more soldiers to this effort than absolutely necessary. His
    generals have marked each square of the map with the number of soldiers required to secure that square.
    What remains is to choose which of those squares should be patrolled.

Write a program to determine the minimum number of soldiers that the emperor will need to be deploy to
    guarantee that the troops of one kingdom cannot move, in one or more steps, into squares occupied by
    the troops of the second kingdom (moving horizontally or vertically) without encountering the emperor's
    own soldiers.

-----Input-----
Input begins with a line containing $2$ integers, $w$ and $h$, denoting the width and height of the map,
    where $1 \leq w, h \leq 40$.

This is followed by $h$ lines. Each line contains $w$ characters, left justified. These characters will be
    'A' or 'B', designating a position held by king A or king B, or a single numeric digit, designating a
    currently unoccupied position that can be secured by the use of that number of soldiers. For example, a
    '2' would indicate that two soldiers must be deployed to that square to secure it against passage of
    other troops. A '0' indicates terrain that is impassable - the emperor need not commit soldiers there
    because the kingdom troops cannot pass through that square.

No 'A' will be adjacent, horizontally or vertically, to any 'B'.

There will be at least one 'A' and one 'B' in the input.

-----Output-----
Print a single line containing an integer denoting the minimum number of soldiers that the emperor must
    deploy to guarantee that there is no open path between any 'A' position and any 'B' position, using any
    combination of horizontal or vertical moves.

-----Examples-----
Sample Input:
8 5
A11111AA
AA7B111A
111BB111
11BBB111
11BBB11B
Sample Output:
13

### INCORRECT PROGRAM
def main():
    w, h = map(int, input().split())
    grid = [input() for _ in range(h)]

    def bfs(x, y, z):
        dist = [[float("inf")] * w for _ in range(h)]
        dist[x][y] = 0
        q = [(x, y)]
        for x, y in q:
            for dx, dy in [(-1, 0), (1, 0), (0, -1), (0, 1)]:
                nx, ny = x + dx, y + dy
                if 0 <= nx < h and 0 <= ny < w and dist[nx][ny] == float("inf"):
                    if grid[nx][ny] == z:
                        dist[nx][ny] = dist[x][y]
                        q.append((nx, ny))
                    if "0" <= grid[nx][ny] <= "9":
                        dist[nx][ny] = dist[x][y] + int(grid[nx][ny])
                        q.append((nx, ny))
        return dist

    ans = float("inf")
    for i in range(h):
        for j in range(w):
```

```
            if grid[i][j] == "A":
                ans = min(ans, min(x for row in bfs(i, j, "B") for x in row))
    print(ans)

main()

### ERROR MESSAGE
===Input===
8 5
A11111AA
AA7B111A
111BB111
11BBB111
11BBB11B
===Program Output===
0
===Expected Output===
13

### FEEDBACK
This code appears to be finding the minimum "distance" between an A position and a B position, where distance
    is measured in the number of soldiers required to secure the intermediate positions. This isn't really
    addressing the question at all, because even occupying the positions along some path from a particular
    A -> B won't necessarily block the kings from reaching each other from those positions through a
    different, longer path. You probably need some sort of min-cut algorithm to divide the graph into two
    connected components, one each for A and B nodes.

### COMMENT
The feedback expresses mild uncertainty in two aspects. First, the user indicates that they are not
    completely certain in their understanding of the code's current behaviour by using the phrase "appears
    to be". Then, they express uncertainty in their suggestion for what to do instead, saying that one
    *probably* needs some sort of min-cut algorithm.
```

# G PROMPTS

In this appendix, we describe the prompting structure used for our experiments. To avoid repetition, we list the prompts used for the GPT models here; for CodeLlama, we use identical prompts *except* that we replace "```python...```" block delimiters with [PYTHON]...[/PYTHON] delimiters, since we find that this significantly improves code extraction reliability.

## G.1 APPS

For initial code generation (the first sample from $M_P$), we use different prompts for the two types of tasks in APPS: call-based tasks, in which the desired program should take the input as a parameter to a function and return the output in the function's return statement; and stdio-based tasks, in which inputs should be read from stdin and outputs should be written to stdout. These prompts are shown in Listing 1 and 2, respectively. The example tasks and programs were taken from APPS' training set.

For feedback samples (i.e., samples from $M_F$), we use the prompt in Listing 3. This prompt contains an example in which the user provides the textual specification, the incorrect program and the error message, and the assistant generates feedback. Similarly, for repair samples (i.e., samples from $M_P$ which follow $M_F$) we use the prompt in Listing 4, in which the user also supplies the feedback, and the assistant returns only the fixed version of the program. Finally, for joint feedback-repair samples (i.e., when sampling $(f, r) \sim M_P$), we use the prompt in Listing 5. This prompt combines the prompts from Listing 3 and 4 into one prompt, in which the assistant returns both the feedback and the fixed program. In all of these prompts, the specification used was taken from APPS' training set, while the programs and the feedback were constructed manually.

Listing 1: Code generation prompt for call-based tasks.

```
=====system=====
You are an expert Python programmer. You will be given a question (problem specification) and will generate a
    correct Python program that matches the specification and passes all tests. You will NOT return
    anything except for the program. Put your fixed program within code delimiters, for example: ```python
# YOUR CODE HERE
```.
=====user=====
### QUESTION
A palindrome is a word, phrase, number, or other sequence of characters which reads the same backward as
    forward. Examples of numerical palindromes are: `2332, 110011, 54322345`

For a given number ```num```, write a function which returns an array of all the numerical palindromes
    contained within each number. The array should be sorted in ascending order and any duplicates should
    be removed.

In this kata, single digit numbers and numbers which start or end with zeros (such as `010` and `00`) are
    **NOT** considered valid numerical palindromes.

If `num` contains no valid palindromes, return `"No palindromes found"`.
Otherwise, return `"Not valid"` if the input is not an integer or is less than `0`.

## Examples

```
palindrome(1221)        =>  [22, 1221]
palindrome(34322122)    =>  [22, 212, 343, 22122]
palindrome(1001331)     =>  [33, 1001, 1331]
palindrome(1294)        =>  "No palindromes found"
palindrome("1221")      =>  "Not valid"
```

---

### Other Kata in this Series:

Numerical Palindrome #1
Numerical Palindrome #1.5
Numerical Palindrome #2
Numerical Palindrome #3
Numerical Palindrome #3.5
Numerical Palindrome #4
Numerical Palindrome #5
### Use Call-Based Format
=====assistant=====
```python
def is_pal(s):
    return s == s[::-1]
```

```
def palindrome(num):
    if not isinstance(num, int) or num < 0:
        return 'Not valid'
    s = str(num)
    pals = set()
    for i, ch in enumerate(s):
        if ch == '0':
            continue
        for j in range(i + 2, len(s) + 1):
            test = s[i:j]
            if is_pal(test):
                pals.add(test)
    return sorted(int(x) for x in pals) or 'No palindromes found'
```

Listing 2: Code generation prompt for stdio-based tasks.

```
=====system=====
You are an expert Python programmer. You will be given a question (problem specification) and will generate a
    correct Python program that matches the specification and passes all tests. You will NOT return
    anything except for the program. Put your fixed program within code delimiters, for example: ```python
# YOUR CODE HERE
```.
=====user=====
### QUESTION
Polycarp has $n$ different binary words. A word called binary if it contains only characters '0' and '1'. For
    example, these words are binary: "0001", "11", "0" and "0011100".

Polycarp wants to offer his set of $n$ binary words to play a game "words". In this game, players name words
    and each next word (starting from the second) must start with the last character of the previous word.
    The first word can be any. For example, these sequence of words can be named during the game: "0101",
    "1", "10", "00", "00001".

Word reversal is the operation of reversing the order of the characters. For example, the word "0111" after
    the reversal becomes "1110", the word "11010" after the reversal becomes "01011".

Probably, Polycarp has such a set of words that there is no way to put them in the order correspondent to the
    game rules. In this situation, he wants to reverse some words from his set so that:  the final set of
    $n$ words still contains different words (i.e. all words are unique);  there is a way to put all words
    of the final set of words in the order so that the final sequence of $n$ words is consistent with the
    game rules.

Polycarp wants to reverse minimal number of words. Please, help him.

-----Input-----

The first line of the input contains one integer $t$ ($1 \le t \le 10\^4$) - the number of test cases in the
    input. Then $t$ test cases follow.

The first line of a test case contains one integer $n$ ($1 \le n \le 2\cdot10^5$) - the number of words in
    the Polycarp's set. Next $n$ lines contain these words. All of $n$ words aren't empty and contains only
    characters '0' and '1'. The sum of word lengths doesn't exceed $4\cdot10^6$. All words are different.

Guaranteed, that the sum of $n$ for all test cases in the input doesn't exceed $2\cdot10^5$. Also, guaranteed
    that the sum of word lengths for all test cases in the input doesn't exceed $4\cdot10^6$.

-----Output-----

Print answer for all of $t$ test cases in the order they appear.

If there is no answer for the test case, print -1. Otherwise, the first line of the output should contain $k$
    ($0 \le k \le n$) - the minimal number of words in the set which should be reversed. The second line of
    the output should contain $k$ distinct integers - the indexes of the words in the set which should be
    reversed. Words are numerated from $1$ to $n$ in the order they appear. If $k=0$ you can skip this line
    (or you can print an empty line). If there are many answers you can print any of them.

-----Example-----
Input
4
4
0001
1000
0011
0111
3
010
101
0
2
00000
00001
4
01
001
0001
00001

Output
```

```
1
3
-1
0

2
1 2
### Use Standard Input Format (read inputs with `input()`, write results with `print()`)
=====assistant=====
```python
for _ in range(int(input())):
    n = int(input())
    mass = []
    zo = 0
    oz = 0
    zz = 0
    oo = 0
    ozs = []
    zos = []
    ozss = set()
    zoss = set()
    for j in range(n):
        k = input()
        mass.append(k)
        if k[0] == '0' and k[-1] == '1':
            zoss.add(k)
            zos.append(j + 1)
            zo += 1
        elif k[0] == '1' and k[-1] == '0':
            ozss.add(k)
            ozs.append(j + 1)
            oz += 1
        elif k[0] == '0' and k[-1] == '0':
            zz += 1
        else:
            oo += 1
    if zz and oo and not oz and not zo:
        print(-1)
        continue
    else:
        if zo > oz:
            print((zo - oz) // 2)
            ans = []
            need = (zo - oz) // 2
            i = 0
            while need:
                zzz = mass[zos[i] - 1][len(mass[zos[i] - 1]) - 1:: -1]
                if zzz not in ozss:
                    ans.append(zos[i])
                    need -= 1
                i += 1
            print(*ans)
        else:
            print((oz - zo) // 2)
            ans = []
            need = (oz - zo) // 2
            i = 0
            while need:
                zzz = mass[ozs[i] - 1][len(mass[ozs[i] - 1]) - 1:: -1]
                if zzz not in zoss:
                    ans.append(ozs[i])
                    need -= 1
                i += 1
            print(*ans)
```
```

Listing 3: Feedback-only prompt.

```
=====system=====
You are a helpful programming assistant and an expert Python programmer. You are helping a user debug a
    program. The user has written some code, but it has some errors and is not passing the tests. You will
    help the user by giving a concise (at most 2-3 sentences) textual explanation of what is wrong with the
    code. You will *not* generate any code, because the user wants to fix the code themselves.
=====user=====
### QUESTION
A palindrome is a word, phrase, number, or other sequence of characters which reads the same backward as
    forward. Examples of numerical palindromes are: `2332, 110011, 54322345`

For a given number ```num```, write a function which returns an array of all the numerical palindromes
    contained within each number. The array should be sorted in ascending order and any duplicates should
    be removed.

In this kata, single digit numbers and numbers which start or end with zeros (such as `010` and `00`) are
    **NOT** considered valid numerical palindromes.

If `num` contains no valid palindromes, return `"No palindromes found"`.
Otherwise, return `"Not valid"` if the input is not an integer or is less than `0`.

## Examples
```

```
```
palindrome(1221)       =>  [22, 1221]
palindrome(34322122)   =>  [22, 212, 343, 22122]
palindrome(1001331)    =>  [33, 1001, 1331]
palindrome(1294)       =>  "No palindromes found"
palindrome("1221")     =>  "Not valid"
```

---

### Other Kata in this Series:

Numerical Palindrome #1
Numerical Palindrome #1.5
Numerical Palindrome #2
Numerical Palindrome #3
Numerical Palindrome #3.5
Numerical Palindrome #4
Numerical Palindrome #5
### Use Call-Based Format
### INCORRECT PYTHON CODE
```python
def is_pal(s):
    return s == s[::-1]

def palindrome(num):
    if not isinstance(num, int) or num < 0:
        return 'Not valid'
    s = str(num)
    pals = set()
    for i, ch in enumerate(s):
        for j in range(i + 2, len(s) + 1):
            test = s[i:j]
            if is_pal(test):
                pals.add(test)
    return sorted(int(x) for x in pals) or 'No palindromes found'
```
The code above is wrong and contains a bug. Given input "1001331" the output was "['0', '33', '1001',
    '1331']" but the expected output was "['33', '1001', '1331']".
=====assistant=====
The following is a concise explanation of the issue: The problem description states that numbers which start
    or end with zeros (such as `010` and `00`) are NOT considered valid numerical palindromes. However, the
    code above does not take this into account and therefore returns `00` as a valid palindrome. This can
    be fixed by checking if the first or last character is `0` before adding the string to the set of
    palindromes.
```

Listing 4: Repair-only prompt.

```
=====system=====
You are a helpful programming assistant and an expert Python programmer. You are helping a user write a
    program to solve a problem. The user has written some code, but it has some errors and is not passing
    the tests. The user has spent some time debugging the program and will provide you with a concise
    textual explanation of what is wrong with the code. You will use this explanation to generate a fixed
    version of the program. Put your fixed program within code delimiters, for example: ```python
# YOUR CODE HERE
```.
=====user=====
### QUESTION
A palindrome is a word, phrase, number, or other sequence of characters which reads the same backward as
    forward. Examples of numerical palindromes are: `2332, 110011, 54322345`

For a given number ```num```, write a function which returns an array of all the numerical palindromes
    contained within each number. The array should be sorted in ascending order and any duplicates should
    be removed.

In this kata, single digit numbers and numbers which start or end with zeros (such as `010` and `00`) are
    **NOT** considered valid numerical palindromes.

If `num` contains no valid palindromes, return `"No palindromes found"`.
Otherwise, return `"Not valid"` if the input is not an integer or is less than `0`.

## Examples

```
palindrome(1221)       =>  [22, 1221]
palindrome(34322122)   =>  [22, 212, 343, 22122]
palindrome(1001331)    =>  [33, 1001, 1331]
palindrome(1294)       =>  "No palindromes found"
palindrome("1221")     =>  "Not valid"
```

---

### Other Kata in this Series:

Numerical Palindrome #1
Numerical Palindrome #1.5
Numerical Palindrome #2
Numerical Palindrome #3
```

```
Numerical Palindrome #3.5
Numerical Palindrome #4
Numerical Palindrome #5
### Use Call-Based Format
### INCORRECT PYTHON CODE
```python
def is_pal(s):
    return s == s[::-1]

def palindrome(num):
    if not isinstance(num, int) or num < 0:
        return 'Not valid'
    s = str(num)
    pals = set()
    for i, ch in enumerate(s):
        for j in range(i + 2, len(s) + 1):
            test = s[i:j]
            if is_pal(test):
                pals.add(test)
    return sorted(int(x) for x in pals) or 'No palindromes found'
```
The code above is wrong and contains a bug. Given input "1001331" the output was "['0', '33', '1001',
    '1331']" but the expected output was "['33', '1001', '1331']".
The following is a concise explanation of the issue: The problem description states that numbers which start
    or end with zeros (such as `010` and `00`) are NOT considered valid numerical palindromes. However, the
    code above does not take this into account and therefore returns `00` as a valid palindrome. This can
    be fixed by checking if the first or last character is `0` before adding the string to the set of
    palindromes.
=====assistant=====
### FIXED PYTHON CODE
```python
def is_pal(s):
    return s == s[::-1]

def palindrome(num):
    if not isinstance(num, int) or num < 0:
        return 'Not valid'
    s = str(num)
    pals = set()
    for i, ch in enumerate(s):
        if ch == '0':
            continue
        for j in range(i + 2, len(s) + 1):
            test = s[i:j]
            if is_pal(test):
                pals.add(test)
    return sorted(int(x) for x in pals) or 'No palindromes found'
```
```

Listing 5: Joint feedback-repair prompt.

```
=====system=====
You are a helpful programming assistant and an expert Python programmer. You are helping a user write a
    program to solve a problem. The user has written some code, but it has some errors and is not passing
    the tests. You will help the user by first giving a concise (at most 2-3 sentences) textual explanation
    of what is wrong with the code. After you have pointed out what is wrong with the code, you will then
    generate a fixed version of the program. Put your fixed program within code ```delimiters, for example:
        ```python
# YOUR CODE HERE
```.
=====user=====
### QUESTION
A palindrome is a word, phrase, number, or other sequence of characters which reads the same backward as
    forward. Examples of numerical palindromes are: `2332, 110011, 54322345`

For a given number ```num```, write a function which returns an array of all the numerical palindromes
    contained within each number. The array should be sorted in ascending order and any duplicates should
    be removed.

In this kata, single digit numbers and numbers which start or end with zeros (such as `010` and `00`) are
    **NOT** considered valid numerical palindromes.

If `num` contains no valid palindromes, return `"No palindromes found"`.
Otherwise, return `"Not valid"` if the input is not an integer or is less than `0`.

## Examples

```
palindrome(1221)      =>  [22, 1221]
palindrome(34322122)  =>  [22, 212, 343, 22122]
palindrome(1001331)   =>  [33, 1001, 1331]
palindrome(1294)      =>  "No palindromes found"
palindrome("1221")    =>  "Not valid"
```

---

### Other Kata in this Series:

Numerical Palindrome #1
```

```
Numerical Palindrome #1.5
Numerical Palindrome #2
Numerical Palindrome #3
Numerical Palindrome #3.5
Numerical Palindrome #4
Numerical Palindrome #5
### Use Call-Based Format
### INCORRECT PYTHON CODE
```python
def is_pal(s):
    return s == s[::-1]

def palindrome(num):
    if not isinstance(num, int) or num < 0:
        return 'Not valid'
    s = str(num)
    pals = set()
    for i, ch in enumerate(s):
        for j in range(i + 2, len(s) + 1):
            test = s[i:j]
            if is_pal(test):
                pals.add(test)
    return sorted(int(x) for x in pals) or 'No palindromes found'
```
The code above is wrong and contains a bug. Given input "1001331" the output was "['0', '33', '1001',
    '1331']" but the expected output was "['33', '1001', '1331']".
=====assistant=====
The following is a concise explanation of the issue: The problem description states that numbers which start
    or end with zeros (such as `010` and `00`) are NOT considered valid numerical palindromes. However, the
    code above does not take this into account and therefore returns `00` as a valid palindrome. This can
    be fixed by checking if the first or last character is `0` before adding the string to the set of
    palindromes.
# The code below is the correct version of the code above, where the issue has been fixed:
### FIXED PYTHON CODE
```python
def is_pal(s):
    return s == s[::-1]

def palindrome(num):
    if not isinstance(num, int) or num < 0:
        return 'Not valid'
    s = str(num)
    pals = set()
    for i, ch in enumerate(s):
        if ch == '0':
            continue
        for j in range(i + 2, len(s) + 1):
            test = s[i:j]
            if is_pal(test):
                pals.add(test)
    return sorted(int(x) for x in pals) or 'No palindromes found'
```
```

## G.2 HUMANEVAL

For HumanEval, since there are no training samples to use for in-context learning, we generate the
initial programs in a zero-shot manner. For feedback, repair, and joint feedback-repair samples we
use the prompts in Listings 6-8. These all include a one-shot example specification, program, error
message and explanation; these were manually crafted based on the specification and gold standard
program for `HumanEval/0`. Although not strictly necessary, we find that using an in-context
example when doing self-repair encouraged the model to use a format in which we could easily
separate (and analyze) the feedback and the repaired program, rather than interleaving the two. Note
that the form of the error message differs slightly from APPS, since we use a different execution
engine for HumanEval.

Listing 6: Feedback-only prompt.

```
=====system=====
You are a helpful programming assistant. You are helping a user write a program to solve a problem. The user
    has written some code, but it has some errors and is not passing the tests. You will help the user by
    giving a detailed but concise textual explanation of what is wrong with the code. You will *not*
    generate any code, because the user wants to fix the code themselves.
=====user=====
### INCORRECT CODE
```python
def approx_unique_elements(numbers: List[float], threshold: float) -> bool:
    """ Check if in given list of numbers, there are no two numbers closer to each other
    than the threshold given.
    >>> approx_unique_elements([1.0, 2.0, 3.0], 0.5)
    True
    >>> approx_unique_elements([1.0, 2.8, 3.0, 4.0, 5.0, 2.0], 0.3)
    False
    """
```

```
    for elem in numbers:
        for elem2 in numbers:
            distance = abs(elem - elem2)
            if distance < threshold:
                return False

    return True
```

The code does not pass the test cases. The error encountered was: `failed: assert
    approx_unique_elements([1.0, 2.0, 3.0], 0.5) == True: False != True`
=====assistant=====
The following is a concise explanation of why the code failed the test: The code fails because it also
    compares elements to themselves. Thus, for any input, the answer will be False. This can be fixed by
    adding a condition to the inner loop that checks that the indices are not the same.

Listing 7: Repair-only prompt.

```
=====system=====
You are a helpful programming assistant and an expert Python programmer. You are helping a user write a
    program to solve a problem. The user has written some code, but it has some errors and is not passing
    the tests. The user has spent some time debugging the program and will provide you with a concise
    textual explanation of what is wrong with the code. You will use this explanation to generate a fixed
    version of the program. Put your fixed program within code delimiters, for example: ```python\n# YOUR
    CODE HERE\n```.
=====user=====
### INCORRECT CODE
```python
def approx_unique_elements(numbers: List[float], threshold: float) -> bool:
    """ Check if in given list of numbers, there are no two numbers closer to each other
    than the threshold given.
    >>> approx_unique_elements([1.0, 2.0, 3.0], 0.5)
    True
    >>> approx_unique_elements([1.0, 2.8, 3.0, 4.0, 5.0, 2.0], 0.3)
    False
    """

    for elem in numbers:
        for elem2 in numbers:
            distance = abs(elem - elem2)
            if distance < threshold:
                return False

    return True
```
The code does not pass the test cases. The error encountered was: `failed: assert
    approx_unique_elements([1.0, 2.0, 3.0], 0.5) == True: False != True`
The following is a concise explanation of why the code failed the test: The code fails because it also
    compares elements to themselves. Thus, for any input, the answer will be False. This can be fixed by
    adding a condition to the inner loop that checks that the indices are not the same.
=====assistant=====
### FIXED CODE
```python
def approx_unique_elements(numbers: List[float], threshold: float) -> bool:
    """ Check if in given list of numbers, there are no two numbers closer to each other
    than the threshold given.
    >>> approx_unique_elements([1.0, 2.0, 3.0], 0.5)
    True
    >>> approx_unique_elements([1.0, 2.8, 3.0, 4.0, 5.0, 2.0], 0.3)
    False
    """

    for idx, elem in enumerate(numbers):
        for idx2, elem2 in enumerate(numbers):
            if idx != idx2:
                distance = abs(elem - elem2)
                if distance < threshold:
                    return False

    return True
```

Listing 8: Joint feedback-repair prompt.

```
=====system=====
You are a helpful programming assistant and an expert Python programmer. You are helping a user write a
    program to solve a problem. The user has written some code, but it has some errors and is not passing
    the tests. You will help the user by first giving a concise (at most 2-3 sentences) textual explanation
    of what is wrong with the code. After you have pointed out what is wrong with the code, you will then
    generate a fixed version of the program. Put your fixed program within code delimiters, for example:
    ```python\n# YOUR CODE HERE\n```.
=====user=====
### INCORRECT CODE
```python
def approx_unique_elements(numbers: List[float], threshold: float) -> bool:
    """ Check if in given list of numbers, there are no two numbers closer to each other
    than the threshold given.
    >>> approx_unique_elements([1.0, 2.0, 3.0], 0.5)
    True
```

```
    >>> approx_unique_elements([1.0, 2.8, 3.0, 4.0, 5.0, 2.0], 0.3)
    False
    """

    for elem in numbers:
        for elem2 in numbers:
            distance = abs(elem - elem2)
            if distance < threshold:
                return False

    return True
```
The code does not pass the test cases. The error encountered was: `failed: assert
    approx_unique_elements([1.0, 2.0, 3.0], 0.5) == True: False != True`
=====assistant=====
The following is a concise explanation of why the code failed the test: The code fails because it also
    compares elements to themselves. Thus, for any input, the answer will be False. This can be fixed by
    adding a condition to the inner loop that checks that the indices are not the same.

### FIXED CODE
```python
def approx_unique_elements(numbers: List[float], threshold: float) -> bool:
    """ Check if in given list of numbers, there are no two numbers closer to each other
    than the threshold given.
    >>> approx_unique_elements([1.0, 2.0, 3.0], 0.5)
    True
    >>> approx_unique_elements([1.0, 2.8, 3.0, 4.0, 5.0, 2.0], 0.3)
    False
    """

    for idx, elem in enumerate(numbers):
        for idx2, elem2 in enumerate(numbers):
            if idx != idx2:
                distance = abs(elem - elem2)
                if distance < threshold:
                    return False

    return True
```

# H  APPS TASKS USED FOR OUR EVALUATIONS

These tasks were randomly sampled from APPS' test set. To avoid distribution shift, we sampled according to the relative frequency of difficulties in the full dataset. We report the resulting list of tasks to aid reproducibility.

| Difficulty | Tasks |
| --- | --- |
| introductory | '4004', '4058', '4063', '4065', '4100', '4108', '4117', '4155', '4164', '4182', '4193', '4195', '4211', '4217', '4241', '4249', '4270', '4275', '4281', '4293', '4333', '4347', '4350', '4356', '4409', '4426', '4431', '4450', '4465', '4484', '4498', '4505', '4507', '4514', '4544', '4553', '4586', '4610', '4662', '4663', '4667', '4677', '4681', '4704', '4716', '4741', '4750', '4786', '4787', '4801', '4855', '4862', '4864', '4870', '4873', '4890', '4897', '4952', '4966', '4984' |
| interview | '0004', '0013', '0033', '0056', '0073', '0074', '0089', '0091', '0124', '0131', '0139', '0162', '0166', '0183', '0186', '0191', '0199', '0205', '0249', '0253', '0268', '0274', '0300', '0304', '0341', '0342', '0413', '0427', '0434', '0466', '0467', '0496', '0501', '0511', '0537', '0564', '0571', '0575', '0579', '0592', '0597', '0626', '0637', '0676', '0704', '0728', '0757', '0765', '0788', '0794', '0804', '0805', '0811', '0829', '0879', '0904', '0915', '0925', '0937', '0948', '0954', '0955', '0972', '0985', '0989', '1018', '1019', '1033', '1046', '1076', '1133', '1140', '1141', '1145', '1146', '1149', '1168', '1185', '1221', '1232', '1256', '1257', '1280', '1285', '1299', '1317', '1347', '1380', '1392', '1393', '1418', '1444', '1448', '1458', '1489', '1517', '1533', '1573', '1635', '1653', '1668', '1672', '1721', '1736', '1748', '1756', '1759', '1775', '1777', '1825', '1850', '1863', '1865', '1870', '1875', '1906', '1917', '1956', '1962', '1967', '1976', '2024', '2049', '2062', '2092', '2093', '2097', '2106', '2172', '2176', '2203', '2231', '2246', '2264', '2266', '2295', '2326', '2328', '2332', '2342', '2361', '2369', '2407', '2408', '2418', '2455', '2463', '2511', '2515', '2516', '2535', '2585', '2623', '2629', '2642', '2651', '2662', '2668', '2673', '2698', '2701', '2709', '2735', '2742', '2752', '2759', '2765', '2787', '2802', '2832', '2835', '2844', '2858', '2885', '2897', '2923', '2932', '2945', '2973', '2980' |
| competition | '3017', '3019', '3054', '3062', '3063', '3066', '3070', '3077', '3083', '3097', '3117', '3135', '3161', '3186', '3209', '3220', '3286', '3287', '3323', '3335', '3353', '3355', '3371', '3375', '3376', '3388', '3404', '3411', '3433', '3441', '3445', '3470', '3481', '3484', '3548', '3557', '3605', '3609', '3634', '3635', '3671', '3679', '3709', '3754', '3769', '3792', '3798', '3799', '3804', '3810', '3819', '3823', '3836', '3843', '3849', '3876', '3913', '3934', '3972', '3974' |

# I  MORE EXAMPLES OF FEEDBACK FROM GPT-4 AND OUR HUMAN PARTICIPANTS

In this appendix, we give five more examples from the study in Section 4.3, so that the keen reader may themself compare and contrast the feedback generated by GPT-4 and by our human participants. Each listing (6-10) contains a textual specification, an incorrect program, an error message, both examples of GPT-4 feedback which were shown to the participants assigned to this task, and the responses from the participants. For each piece of feedback, we also list the number of resulting repaired programs which passed all the tests (c.f. the full results table in Appendix E).

Listing 9: Task 2106, Program A.

```
### SPECIFICATION
There are n cities in the country where the Old Peykan lives. These cities are located on a straight line,
    we'll denote them from left to right as c_1, c_2, ..., c_{n}. The Old Peykan wants to travel from city
    c_1 to c_{n} using roads. There are (n - 1) one way roads, the i-th road goes from city c_{i} to city
    c_{i} + 1 and is d_{i} kilometers long.

The Old Peykan travels 1 kilometer in 1 hour and consumes 1 liter of fuel during this time.

Each city c_{i} (except for the last city c_{n}) has a supply of s_{i} liters of fuel which immediately
    transfers to the Old Peykan if it passes the city or stays in it. This supply refreshes instantly k
    hours after it transfers. The Old Peykan can stay in a city for a while and fill its fuel tank many
    times.

Initially (at time zero) the Old Peykan is at city c_1 and s_1 liters of fuel is transferred to it's empty
    tank from c_1's supply. The Old Peykan's fuel tank capacity is unlimited. Old Peykan can not continue
    its travel if its tank is emptied strictly between two cities.

Find the minimum time the Old Peykan needs to reach city c_{n}.

-----Input-----

The first line of the input contains two space-separated integers m and k (1 <= m, k <= 1000). The value m
    specifies the number of roads between cities which is equal to n - 1.

The next line contains m space-separated integers d_1, d_2, ..., d_{m} (1 <= d_{i} <= 1000) and the following
    line contains m space-separated integers s_1, s_2, ..., s_{m} (1 <= s_{i} <= 1000).

-----Output-----

In the only line of the output print a single integer - the minimum time required for The Old Peykan to reach
    city c_{n} from city c_1.

-----Examples-----
Input
4 6
1 2 5 2
2 3 3 4

Output
10

Input
2 3
5 6
5 5

Output
14

### INCORRECT PROGRAM
def main():
    m, k = map(int, input().split())
    ds = list(map(int, input().split()))
    ss = list(map(int, input().split()))

    time = 0
    fuel = ss[0]
    min_time = 0
    max_s = ss[0]

    for i in range(m):
        fuel -= ds[i]
        max_s = max(max_s, ss[i])
        while fuel < 0:
            time += k
            fuel += max_s
        time += ds[i]

    print(time)

main()
```

```
### ERROR MESSAGE
===Input===
4 6
1 2 5 2
2 3 3 4

===Program Output===
28

===Expected Output===
10

### GPT-4 FEEDBACK #1
The issue with the code is that it does not consider the time spent in the current city before moving to the
    next city, resulting in an overestimation of the minimum time required. This can be fixed by updating
    the time spent in the current city before calculating the fuel used and accounting for the time it
    takes to travel to the next city.
# Number of passing repairs: 7/25

### GPT-4 FEEDBACK #2
The following is a concise explanation of the issue: The code incorrectly calculates the waiting time at each
    city. It should minimize the waiting time by considering the maximum fuel supply that can be collected
    during the waiting time. The code needs to be updated to account for this.
# Number of passing repairs: 10/25

### PARTICIPANT FEEDBACK #1
The program doesn't add the fuel for passing through cities, only if it needs to actually stay in the city to
    refuel. Whenever it passes through a city, it should add ss[i] to the fuel.
# Number of passing repairs: 10/25

### PARTICIPANT FEEDBACK #2
The code is incorrect because it doesn't take into account the fuel at each city that is immediately
    transferred after entering the city (aside for the fuel received at the first city). As a result, this
    program overestimates the time needed, because it assumes that the Old Peykan always has to wait for a
    refill (besides the fuel they are given in the first city).
# Number of passing repairs: 0/25
```

Listing 10: Task 3070, Program B.

```
### SPECIFICATION
In the free-market, ruthlessly capitalist world of train fares, only one thing matters: incentives.

Train companies are incentivised with bonuses for high throughput, successful journeys, and customer
    satisfaction. Conversely, the companies are disincentivised from failure via mandatory refunds for
    customers delayed by $30$ minutes or more.

Being a ruthless capitalist yourself, you have decided to take advantage of this generous delay compensation
    provision.

The refund is awarded provided that no matter the combination of trains you had taken (provided they followed
    the same route of stations as planned), you would still be unable to reach your destination in strictly
    less time than $30$ minutes (or $1800$ seconds), of the time you would have arrived assuming your
    booked journey was exactly on time.

Armed with your printout of the day's delays, and the original timetable, you must ask yourself only one
    question: what is the earliest time you can book a train for from station $1$, in order to earn this
    restitutive reward?

-----Input-----
- One line containing two integers: $N$ ($1 \le N \le 100$), the number of stations, and $M$ ($1 \le M \le
    10^5$), the number of scheduled trains.
- The next $M$ lines each contain 4 integers:
- $X$, the starting station ($1 \le X \le N-1$),
- $S$ and $T$ ($0 \le S \le T < 86400$), the planned departure and arrival times in seconds,
- and $L$ ($0 \le L < 86400$), the duration by which the train's departure and arrival times are delayed.

Stations are numbered from $1$ to $N$ in the order you will visit them. Each train goes between stations $X$
    and $X+1$. It is possible to change between trains instantanesouly.

-----Output-----
- One line containing one integer: the start time of the earliest train journey you could book in order to
    earn your compensation, or impossible if no such journey is possible.

-----Examples-----
Sample Input 1:
2 3
1 1800 9000 1800
1 2000 9200 1600
1 2200 9400 1400
Sample Output 1:
1800

Sample Input 2:
2 2
1 1800 3600 1800
1 1900 3600 1600
Sample Output 2:
impossible
```

```
### INCORRECT PROGRAM
def main():
    N, M = map(int, input().split())
    train_data = [tuple(map(int, input().split())) for _ in range(M)]

    train_data.sort(key=lambda x: (x[0], x[1]))
    times = [1e9] * (N + 1)
    times[1] = 0

    for X, S, T, L in train_data:
        if times[X] <= S:
            times[X + 1] = min(times[X + 1], T + L)

    if times[N] >= 1800:
        print(int(times[1]))
    else:
        print("impossible")

main()

### ERROR MESSAGE
===Input===
2 3
1 1800 9000 1800
1 2000 9200 1600
1 2200 9400 1400
===Program Output===
0
===Expected Output===
1800

### GPT-4 FEEDBACK #1
The issue with the code is that it prints `times[1]` instead of the earliest possible train start time. The
    code needs to get the minimum start time from the train_data list that has a valid delay for
    compensation.
# Number of passing repairs: 3/25

### GPT-4 FEEDBACK #2
The issue with the code is that it directly prints the value of `times[1]`, which is initialized to `0` and
    never updated. The correct output should be the starting time of the earliest train journey that
    satisfies the condition. To fix this, we need to track the earliest possible start time and update it
    if a valid train journey is found.
# Number of passing repairs: 0/25

### PARTICIPANT FEEDBACK #1
This code is clearly wrong as times[1] is initialized to 0 then never updated and it is returned as the
    answer. Also, in the sorting line `int(x[0])` should be used in place of `x[0]` and likewise for
    `int(x[1])` in place of `x1` because we want numerical order sorting not string order sorting. The goal
    in this problem is calculate for each station what the earliest-arriving undelayed path to the station
    is and what the earliest-arriving delayed path to the station is, take the difference between the two,
    and see if it results in a total delay of >= 30 minutes (then to return the earliest departure from
    station 1 for this path. Given this, first the current code based around `times` can remain - this is
    the delayed time. However, the if-statement within the for-loop should be `<= S + L` since the
    departure time is also delayed. Then `undelayed_times` should be tracked as well, updated in the foor
    loop when `undelayed_times[X] <= S` and set to a min() between the current undelayed time an `T` (not
    T+L). Finally the if-statement for time printing can use the difference between the delayed and
    undelayed times and see if any() are >= 1800. Finally we need to print the earliest arriving train for
    that path
# Number of passing repairs: 5/25

### PARTICIPANT FEEDBACK #2
One of the main issues is that the code prints 'times[1]', which is set to 0 and never updated. However,
    there are also some larger issues with the code. First, instead of checking if you reach your
    destination within 30 minutes of the expected time, it checks if you reach your destination in 30
    minutes. To fix this, the program would need to keep track of the time you'd reach your destination if
    there were not delays, which it currently doesn't. Second, instead of printing times[1], it should be
    keeping track of the start time of the train that leads to a delay and printing this.
# Number of passing repairs: 0/25
```

Listing 11: Task 4333, Program A.

```
### SPECIFICATION
There is a square in the xy-plane. The coordinates of its four vertices are (x_1,y_1),(x_2,y_2),(x_3,y_3) and
     (x_4,y_4) in counter-clockwise order.
(Assume that the positive x-axis points right, and the positive y-axis points up.)
Takahashi remembers (x_1,y_1) and (x_2,y_2), but he has forgot (x_3,y_3) and (x_4,y_4).
Given x_1,x_2,y_1,y_2, restore x_3,y_3,x_4,y_4. It can be shown that x_3,y_3,x_4 and y_4 uniquely exist and
     have integer values.

-----Constraints-----
- |x_1|,|y_1|,|x_2|,|y_2| \leq 100
- (x_1,y_1) != (x_2,y_2)
- All values in input are integers.

-----Input-----
Input is given from Standard Input in the following format:
x_1 y_1 x_2 y_2

-----Output-----
Print x_3,y_3,x_4 and y_4 as integers, in this order.
```

```
-----Sample Input-----
0 0 0 1

-----Sample Output-----
-1 1 -1 0

(0,0),(0,1),(-1,1),(-1,0) is the four vertices of a square in counter-clockwise order.
Note that (x_3,y_3)=(1,1),(x_4,y_4)=(1,0) is not accepted, as the vertices are in clockwise order.

### INCORRECT PROGRAM
x1, y1, x2, y2 = map(int, input().split())

x3 = x2 + (y2 - y1)
y3 = y2 - (x2 - x1)
x4 = x1 + (y2 - y1)
y4 = y1 - (x2 - x1)

print(x3, y3, x4, y4)

### ERROR MESSAGE
===Input===
0 0 0 1

===Program Output===
1 1 1 0

===Expected Output===
-1 1 -1 0

### GPT-4 FEEDBACK #1
The issue with the code is that it doesn't consider the counter-clockwise order of the vertices. To fix this,
    we need to swap the signs of (y2 - y1) and (x2 - x1) in the calculations.
# Number of passing repairs: 25/25

### GPT-4 FEEDBACK #2
The following is a concise explanation of the issue: The code above does not ensure that the vertices are in
    counter-clockwise order. It calculates x3, y3, x4, and y4 in the wrong order, which results in the
    coordinates being in clockwise order instead. To fix this, we need to swap the calculations for x3 and
    x4, and y3 and y4.
# Number of passing repairs: 0/25

### PARTICIPANT FEEDBACK #1
The program calculates vertices for a clockwise-oriented square, not counter-clockwise. To fix this, (y2 -
    y1) and (x2 - x1) should be negated in the equations.
# Number of passing repairs: 25/25

### PARTICIPANT FEEDBACK #2
The calculation for y3 and y4 incorrectly subtract the x direction vector to y0 and y1 respectively. In each
    of these cases, the calculation can be fixed by changing the first subtraction to addition.
# Number of passing repairs: 0/25
```

Listing 12: Task 4347, Program B.

```
### SPECIFICATION
One day, $n$ people ($n$ is an even number) met on a plaza and made two round dances, each round dance
    consists of exactly $\frac{n}{2}$ people. Your task is to find the number of ways $n$ people can make
    two round dances if each round dance consists of exactly $\frac{n}{2}$ people. Each person should
    belong to exactly one of these two round dances.

Round dance is a dance circle consisting of $1$ or more people. Two round dances are indistinguishable
    (equal) if one can be transformed to another by choosing the first participant. For example, round
    dances $[1, 3, 4, 2]$, $[4, 2, 1, 3]$ and $[2, 1, 3, 4]$ are indistinguishable.

For example, if $n=2$ then the number of ways is $1$: one round dance consists of the first person and the
    second one of the second person.

For example, if $n=4$ then the number of ways is $3$. Possible options: one round dance - $[1,2]$, another -
    $[3,4]$; one round dance - $[2,4]$, another - $[3,1]$; one round dance - $[4,1]$, another - $[3,2]$.

Your task is to find the number of ways $n$ people can make two round dances if each round dance consists of
    exactly $\frac{n}{2}$ people.

-----Input-----

The input contains one integer $n$ ($2 \le n \le 20$), $n$ is an even number.

-----Output-----

Print one integer - the number of ways to make two round dances. It is guaranteed that the answer fits in the
    $64$-bit integer data type.

-----Examples-----
Input
2

Output
1
```

```
Input
4

Output
3

Input
8

Output
1260

Input
20

Output
12164510040883200

### INCORRECT PROGRAM
def factorial(n):
    if n == 0:
        return 1
    else:
        return n * factorial(n-1)

n = int(input())

ans = (factorial(n) // (n // 2)**2) // (2 * n)
print(ans)

### ERROR MESSAGE
===Input===
2

===Program Output===
0

===Expected Output===
1

### GPT-4 FEEDBACK #1
The issue with the code is that it uses integer division when calculating the final answer, which may cause
    the result to be rounded down incorrectly. To fix this, we can use float division and then convert the
    result to an integer.
# Number of passing repairs: 0/25

### GPT-4 FEEDBACK #2
The following is a concise explanation of the issue: The code attempts to calculate the number of ways using
    integer division, which may result in a loss of precision and incorrect results. To fix the issue,
    change integer division to regular division, then round the result to the nearest integer using
    `round()`.
# Number of passing repairs: 0/25

### PARTICIPANT FEEDBACK #1
The problem is that the formula used is incorrect. For n <= 4, n! is less than n^3 / 2 and so this is why we
    see a 0. There is an erroneous additional division by n which is the true reason for the wrong answer.
# Number of passing repairs: 25/25

### PARTICIPANT FEEDBACK #2
On line 11, the program calculates a term with a correct numerator, but an incorrect denominator. It divides
    by (2*n), when it should just divide by 2.
# Number of passing repairs: 25/25
```

Listing 13: Task 4855, Program A.

```
### SPECIFICATION
The impact factor of a scientific journal is a measure reflecting the average number of citations to articles
    published in science journals. For this task we are using a simplified formula for calculating the
    impact factor:\[ \frac{\text {Total count of citations received by articles published in the
    journal}}{\text {Total number of articles published}} \]

Rounding is always performed upwards. For example the impact factor of the "Journal for ore research and time
    wasting" that published $38$ articles cited $894$ times is $894 / 38 = 23.53$ rounding up to $24$.

You are the editor of one scientific journal. You know how many articles you are going to publish and the
    owners are pushing you to reach a specific impact factor. You are wondering how many scientists you
    will have to bribe to cite your articles to meet the owners demands. Since money is tight you want to
    bribe the minimal amount of scientists. Each bribed scientist buys you a single citation.

-----Input-----
First and only line of input will contain $2$ integers, $A$ ($1 \leq A \leq 100$), the number of articles you
    plan to publish and $I$ ($1 \leq I \leq 100$), the impact factor the owners require.

-----Output-----
The first and only line of output should contain one integer, the minimal number of scientists you need to
    bribe.

-----Examples-----
Sample Input 1:
```

```
38 24
Sample Output 1:
875

Sample Input 2:
1 100
Sample Output 2:
100

### INCORRECT PROGRAM
import math

A, I = map(int, input().split())
needed_citations = math.ceil(A * (I - 0.5))
scientists_to_bribe = needed_citations - A * (I - 1)
print(scientists_to_bribe)

### ERROR MESSAGE
===Input===
38 24

===Program Output===
19

===Expected Output===
875

### GPT-4 FEEDBACK #1
The following is a concise explanation of the issue: The calculation for the number of scientists to bribe is
    incorrect. It should be calculated by first finding the minimum number of citations needed: `A * I`,
    and then subtracting the current number of citations: `A * (I - 1)`.
# Number of passing repairs: 0/25

### GPT-4 FEEDBACK #2
The following is a concise explanation of the issue: The code calculates the required number of scientists to
    bribe by subtracting the current number of citations from the number of needed citations. However, it
    incorrectly calculates the current number of citations.
# Number of passing repairs: 1/25

### PARTICIPANT FEEDBACK #1
The scientists_to_bribe variable on line 3 is unnecessary., It's sufficient to just do A * (I - 1) + 1.
# Number of passing repairs: 17/25

### PARTICIPANT FEEDBACK #2
The program seems to have a conceptual misunderstanding. Specifically, the number of scientists needed to
    bribe is equal to the number of required citations (`citations_needed`). However, the calculation of
    `citation_needed` is also incorrect -- it implies rounding impact factor conventionally (e.g. 3.49 goes
    to 3, 3.51 goes to 4), whereas rounding here is a ceiling function on the impact factor. As such, the
    actual formula for `citation_needed` should be `A*(I-1) + 1`
# Number of passing repairs: 25/25
```

