# OpenReview forum: "Is Self-Repair a Silver Bullet for Code Generation?"
_ICLR.cc/2024/Conference — ICLR 2024 poster_

### Official Review · Reviewer_LPE6 · 2023-10-31

**Soundness:** 3 good
**Presentation:** 3 good
**Contribution:** 4 excellent
**Rating:** 8
**Confidence:** 4

**Summary:**

The authors break the problem of program repair into four steps:

1. (LLM) Generate an initial population of programs.
2. Run the programs to obtain error messages.
3. (LLM) Analyze the errors, and generate feedback as to the cause of failure.
4. (LLM) Given the feedback, generate a repaired program.

This paper differs from prior work in part because it emphasizes the importance of step (3) -- using feedback to direct the repair.  The authors run a number of experiments with different models, explore what happens when a stronger model provides feedback to a weaker one, and also explore what happens when humans provide feedback to a model.

The authors also explicitly model the cost of repair.  Is running steps (3) and (4) more effective than simply drawing more samples from the initial population?  The answer turns out to be "not really".

The experiments are well-designed and the authors make several interesting observations.  To paraphrase:

A.  LLMs are currently quite bad at program repair.  The best way to get a correct program is to generate a large and diverse initial population, in the hopes that one of the initial programs is "close enough" to work with minimal tweaks.  Even a single repair step yields at best marginal improvements.  LLMs are thus very different from human programmers; they seem to be unable to iteratively refine and debug programs over multiple steps.

B.  The main limitation is the quality of feedback.  Given feedback from a stronger model, or from a human, LLMs show substantial gains from repair.  However, LLMs seem to have difficulty finding problems in the code that they generate.

**Strengths:**

The paper is well-written, and the authors formulation of the problem is insightful.  In particular, they do not merely test the accuracy of program repair in isolation, they compare it against the alternative, which is generating new initial programs from scratch.

The experiments seem to be well done, particularly the ones which use a stronger model or human to provide feedback to a weaker model.

**Weaknesses:**

The authors do not attempt to fine-tune a model on the task of program repair.  Thus their experiments mainly demonstrate that LLMs, *as currently trained*, do not do a good job at the repair task.

To be fair, fine-tuning is probably out of scope for this paper, especially since some of they models they test are private, and accessible only via an API.

**Questions:**

Your illustration of the repair-tree is interesting, and it brings to mind the idea of extending this evaluation technique to a proper tree search.  You might be able to get better results by using a value network, and focusing only on the nodes of the tree that are most promising for repair, in a manner reminiscent of evolutionary search, with the LLM as mutator.

Successful repair attempts, preferably after several iterations of feedback and repair, could also be used to fine-tune the LLM to generate better feedback, and to generate better repair code given the feedback.  (See weaknesses, above.)

Have you considered any experiments along these lines?

---

> ### Author Response · Authors · 2023-11-14
> **Response to Reviewer LPE6**
>
> Thank you, Reviewer LPE6, for your kind words and suggestions for further experiments! We are happy to hear that you find the paper well written, and the methodology/experiment design both sound and insightful.
>
> As also pointed out by Reviewer PNGf01, you are correct that our contributions are limited to the few-shot/in-context setting. We share your sentiment that it would be very interesting to see how finetuning would affect the conclusions in this paper, but—as you have pointed out—this is out of scope for our experiments (which we emphasize are already very large, due to the branching nature of self-repair). We look forward to future work which can leverage the insights we share here to investigate what role fine-tuning can play in alleviating self-repair’s feedback bottleneck!
>
> As to your questions: You touch on two important ideas, that we certainly share your excitement about.
>
> The first is to use a value network to select which candidate program to repair at each step. Our preliminary experiments did explore some very simple search strategies, such as ranking the candidates by their cumulative log probabilities, but we found no evidence that this was correlated positively with self-repair success rate. However, it is very much possible that using a value network/“confidence model” to rank the candidates (Chen et al., 2021; Inala et al., 2022; Zhang et al., 2022; see Related Work) would yield performance benefits. Although this fell outside of the scope of this paper, we are excited about future work investigating when and how proper tree search can be used in conjunction with self-repair to yield greater performance increases.
>
> Your second question is reminiscent of an RLHF (without the H) training stage, in which the model uses self-sampled feedback (and repairs) to improve its own code debugging & repair capabilities. This is certainly an interesting idea, and one we believe will play an important role in the future as new (publically available, permissively licensed) datasets to train on become more rare. However, it is also a method with clear challenges, and not one we have started experimenting with at this stage!
>
> Thanks again for your feedback. Let us know if you have any further questions; we'd love to continue this discussion through the rebuttal week and beyond!

---

> > ### Comment · Reviewer_LPE6 · 2023-11-17
> >
> > Thank you for your response.  I'm glad to hear that you are thinking about value networks and RL(-H)F; I look forward to seeing further work in this space.

---

### Official Review · Reviewer_PNGf · 2023-11-02

**Soundness:** 3 good
**Presentation:** 3 good
**Contribution:** 3 good
**Rating:** 8
**Confidence:** 4

**Summary:**

This paper investigates the ability of large language models (specifically, Code Llama, GPT-3.5, and GPT-4) to perform self-repair which refers to the model's capacity to identify and correct mistakes in its own generated code. The paper analyzes the effectiveness of self-repair in addressing code generation errors, specifically on problems sourced from HumanEval or APPS datasets. The findings suggest that the gains achieved through self-repair are often modest and vary significantly across different subsets of data. In some cases, self-repair does not result in noticeable improvements. The paper proposes that this limitation might be due to the model's ability to provide feedback on its own code, and stronger models might enhance self-repair effectiveness. Additionally, the paper explores the impact of providing the model with feedback from human participants, showing that this feedback significantly benefits the self-repair process, even for the advanced GPT-4 model. The paper offers a brief qualitative analysis to shed light on why this human feedback is valuable in improving code repair performance. Overall, the study provides insights into the challenges and potential enhancements of self-repair mechanisms in large language models for code generation tasks.

**Strengths:**

__The paper systematically analyzes the self-repair capability of LLMs.__

While existing works of self-repair mostly argue that self-repair is feasible for LLM, this paper is the first work to systematically study the strengths and weaknesses of self-repair. Specifically, I appreciate the systematic comparison between self-repair and the pass@k performance, though the conclusion might not be generalizable (see Weaknesses), showing that when the computation is comparable, their benefits are also similar, and also suggesting the optimal combination of these two techniques is the most effective. Also, studying the quality of feedback is also a novel perspective to analyze the self-repair capability of LLM

__The paper sheds light on the weaknesses of LLM in self-repair, providing clear takeaways and indicating the potential future work in this direction.__

The paper is well written and properly organized, and it sheds light on the weaknesses of LLM's self-repair capability. Such weaknesses can be due to the lack of both data and carefully crafted training strategy for self-repair capability, indicating the future research direction of refining the existing LLM with better self-repair capability.

**Weaknesses:**

I am overall positive regarding this paper, and I appreciate the systematic study the paper performs to quantify LLM's self-repair capability. However, the conclusions are claimed in a strong and general tone, while the study itself is actually limited in scope for two reasons.

__The scope of the study is limited to solving isolated programming challenges while ignoring real-world development.__

The study focuses completely on the programming challenges datasets, such as HumanEval and APPs. These datasets have several characteristics that are not realistic in daily development, therefore, though I appreciate the initial conclusions of this paper, these takeaways might not be applicable to a more realistic scenario.

First, the samples in the studied datasets are provided with clear and complete problem descriptions, which are not always available in real-world programming practice. One of the main reasons that pass@k works so well in programming challenges is that the expected functionality of the program is fully revealed and clearly explained in the prompt, so the model is able to maximize the diversity within a narrowed semantic space when generating multiple sequences. However, such clearly explained prompts, as natural language, are typically not available during the ongoing development, where the developers start with a very high-level goal and eventually design modules and implement them piece by piece. During this process, the human intent is not always explicitly specified, as docstring or comment, before the code LM is prompted to complete the following code snippets. In these cases, the execution of unit tests provide meaningful feedback to specify and concretize the expected functionalities, which cannot be leveraged by the top-k generation but is valuable guidance for iterative self-repair. Therefore, though I agree with the takeaway that pass@k is comparable to, sometimes better than, self-repair in the programming challenge dataset, such observation might not be realistic for daily development and requires further study.

Second the samples in the studied datasets are mostly short and self-contained, missing the complicated data and inter-procedural dependencies. Pass@k explores the breath of each token within the sequence without directional guidance, but such breath or search space exponentially increases with the increase of the code length. The program challenges datasets contain samples mostly up to tens of lines of code, significantly underestimating the complexity of real-world software, which includes hundreds or thousands of lines of code within one single file and maintains complicated dependencies. Generating k sequences blindly without feedback may hardly fulfill the expectation due to the large search space, while execution feedback, such as the indication of a missing third-party library, helps the model quickly locate the problematic code and focus on fixing just that part. Therefore, when the complexity of the program increases, it requires further study to understand whether self-repair is equivalent as top-k generation.

__It is not clear whether fine-tuning for self-repair could easily overcome the weaknesses or not.__

This paper focuses on the LLM's self-repair capability only with prompting, without optimizing the model parameters towards the self-repair capability. It is not clear whether a cheap fine-tuning could quickly enable the model's capability of understanding and leveraging the feedback efficiently. Drawing the conclusion that self-repair is not a silver bullet without trying straightforward fine-tuning might be too strong.

To conclude, I would encourage the author to consider constraining their conclusions with the study's scope and add a discussion section to mention

**Questions:**

Please explain and address the weaknesses. Otherwise, the paper is well-written and clear.

---

> ### Author Response · Authors · 2023-11-14
> **Response to PNGf**
>
> We thank the reviewer for their feedback on the limitations of our analysis! We agree that there is much left to explore in this space, and are excited about the future work this work will inspire. Without further ado, let’s jump into a discussion of the weaknesses identified by PNGf01.
>
> *The scope of our study is limited to self-contained Python programs, which is quite different from real-world software engineering.*
>
> Software Engineering (SE) and competitive programming both present their own unique challenges. SE tasks often involve incomplete  task specifications as well as (long) contextual dependencies. Competitive programming, on the other hand, is mainly challenging due to the logical and algorithmic complexity, often requiring the use of dynamic programming and graph algorithms to solve the task.
>
> An ideal self-repair model should be able to (1) fix errors in logically complex tasks, (2) handle ambiguous specifications and missing context, and (3) repair without a clear test oracle. The community would benefit from in-depth studies of each capability. In this paper, we isolate the first aspect, providing insights that would be hard to untangle otherwise.
>
> With this in mind, we believe competitive programming becomes a well suited testbed for our analysis:
> - Using logically complex, well-specified programming puzzles to benchmark models has a rich history in the literature [0,1,2].
> - Recent work shows that even contemporary models are still challenged by intermediate/advanced-level competitive programming tasks [3, 4].
> - Competitive programming tasks have relatively complete specifications and unit tests, so we do not have to worry as much about our results being muddled by degenerate cases (e.g. the solution is incorrect simply because the task specification did not provide a sufficient definition of an API).
>
> In our study, we are thus able to hone in on self-repair's efficacy in algorithmically challenging tasks of varying difficulty levels. We use this primarily to isolate the importance of the feedback stage, but it also enables us to discover non-obvious relationships such as the non-straightforward interplay between task difficulty and self-repair efficacy (Section 4.1 and Appendix B). This surprising fact would have been difficult to discover without isolating the effect of the logical/algorithmic complexity of the task at hand.
>
> In future work, we'd like to explore the other two capabilities as well. These are motivated by the question of how future software engineering workflows should best leverage AI code generation tools. One can imagine workflows which emphasize encapsulation to an extent that each individual piece is no more complex than a programming puzzle, but tricky bugs still tend to arise at the interface level. Furthermore, the fact that Test-Driven Development has fallen out of vogue suggests that developers do not like writing unit tests first, which motivates capability (3). Similarly, although natural language may not have played a big role in software engineering historically, minimizing the impact of ambiguity in NL specifications is now necessary in order to enable capability (2).
>
> In summary, as we hope this discussion has shown, we wholeheartedly agree with the reviewer that there is much interesting work left to be done in this space. Furthermore, **we will revise the Introduction and Future Work sections to clarify the scope of our contributions, as well as highlight some of the nuances discussed above**.
>
> *Our study is limited to prompting strategies, and does not consider fine-tuning.*
>
> We certainly agree that such an analysis would be of interest to the community! However, we believe that this falls outside the scope of this study, which we emphasize is already significant in terms of its depth (and cost). In addition to the methodological challenges, fine-tuning also comes with practical concerns such as cost and model availability. We thus leave it to future work to investigate whether fine-tuning can alleviate the feedback bottleneck identified by this paper. Besides, prompting may be accessible to a wider audience than fine-tuned models in the current AI ecosystem; studying self-repair in this context may therefore benefit AI practitioners more in practice.
>
> Please let us know if you have any further thoughts or questions - we greatly appreciate your feedback!
>
>
> ---
>
>
> [0] Li, Yujia, et al. "Competition-level code generation with alphacode." Science 378.6624 (2022): 1092-1097.
>
> [1] Austin, Jacob, et al. "Program synthesis with large language models." arXiv preprint arXiv:2108.07732 (2021).
>
> [2] Chen, Mark, et al. "Evaluating large language models trained on code." arXiv preprint arXiv:2107.03374 (2021).
>
> [3] Hendrycks, Dan, et al. "Measuring coding challenge competence with APPS." arXiv preprint arXiv:2105.09938 (2021).
>
> [4] Inala, Jeevana Priya, et al. "Fault-aware neural code rankers." Advances in Neural Information Processing Systems 35 (2022): 13419-13432.

---

> > ### Comment · Reviewer_PNGf · 2023-11-22
> >
> > Thanks for the authors' revision of the paper and thoughtful discussion. I am now increasing my score to champion this paper for its acceptance.

---

> > > ### Author Response · Authors · 2023-11-22
> > > **Re: recommendation of acceptance**
> > >
> > > Thank you PNGf for your feedback! We are very pleased to hear that we were able to address your concerns in the discussion as well as the revised version of the paper, and that you now feel confident recommending that the paper be accepted to the conference. Thanks again! :)

---

### Official Review · Reviewer_Vfe4 · 2023-11-07

**Soundness:** 3 good
**Presentation:** 3 good
**Contribution:** 3 good
**Rating:** 6
**Confidence:** 3

**Summary:**

This paper investigates the sample efficiency of a self-repair approach for LLM based code-generation tasks. It evaluates performance of this approach on HumanEval and APPS dataset using ColdeLLama-13b-instruct, GPT3.5, and GPT4  and provides several insights based on the results: 1) Sampling without repair can perform equally or better than self-repair in almost all sample budgets; 2) Initial sampling diversity is more critical than the diversity of repair samples; 3) Quality of feedback significantly improves the performance of self-repair.

**Strengths:**

* This paper provides several new insights on self-repair for code generation compared to the baseline method of sample generations without self-repair. It shows how sample budget and initial sample diversity could impact the efficiency of code generation.
* The investigations on self-repair performance improvement only by improving feedback quality could enable more interesting future ideas.
* Overall, the paper is well-written and easy to read. The authors did a great job in highlighting the key limitations of the analysis.

**Weaknesses:**

The experimental results presented in this support the claim around the limitations of self-repair. Interestingly, the findings around overall efficacy compared to baseline somewhat contradicts with the results from Chen et. al. 2023b that shows self-repair could provide significant increase in sample efficiency. Although ‘self-debuggging’ work from Chen et. al. is mentioned in the related work, I think more comparative analysis would strengthen the claim of this paper. Analysis results from more diverse code generation task including datasets other than python language would also be interesting additions to the analysis.

**Questions:**

1. Is there a specific reason to restrict feedback and repair samples to 1 in the analysis of feedback boosting (section 4.2)?
2. Should we expect similar results using the ‘self-debugging’ approach that uses few-shot prompts?

---

> ### Author Response · Authors · 2023-11-14
> **Response to Vfe407**
>
> We thank Reviewer Vfe407 for their helpful comments! We are very happy to hear that the reviewer found our contributions insightful, easy to understand and well scoped (with clear discussion of limitations).
>
> We agree with the reviewer that the relationship to Chen et al. (2023b) is important to discuss. After replying to the reviewer’s specific questions below, we will give a detailed account of how our work compares to that of Chen et al (2023b).
>
> ---
>
> *Individual responses to questions.*
>
> > Is there a specific reason to restrict feedback and repair samples to 1 in the analysis of feedback boosting (section 4.2)?
>
> This restriction is mainly a practical one: separating the feedback and repair makes the experiment significantly more time-consuming (and costly) to run, since each stage must now be implemented as a separate API call. With this restriction, we can control the cost of this experiment by using a smaller value of $N_f$ and $N_r$ without increasing the risk of statistical artifacts too much. We emphasize that the preceding experiments already showed that this setting is the most effective, and we therefore do not believe this limits the analysis in practice.
>
> > Should we expect similar results using the ‘self-debugging’ approach that uses few-shot prompts?
>
> Although our results are already similar (and we do use few-shot prompting; see Appendix F for a complete list of the prompts we use), we agree that our results are–generally speaking–not as strongly in favor of self-repair/self-debugging. This is because of slight differences in the experimental setting being investigated; see the discussion below for a detailed comparison.
>
> ---
>
> *Detailed comparison to Chen et al. (2023b).*
>
> As mentioned in the Related Work, Chen et al. (2023b)’s method is indeed closely related to ours; both use few-shot prompting to encourage the model to first retrospect on the code (in order to understand why it failed) and then perform repair. There are, however, a few differences.
>
> 1. The main reason our results differ from those of Chen et al. (2023b) is that we conduct our experiments in a slightly different setting.
>
> In Chen et al.’s study, the self-debugging method has access to a correctness oracle to decide whether to proceed with debugging or not; the baseline does not have access to the oracle. In our work, both the baseline and the self-repair approach have equal access to an oracle (when evaluating pass@k) and are compared with the same sample budget. Thus, the performance improvement over the baseline is more prominent in Chen et al. (2023b)’s work than in ours.
>
> We choose to grant both baseline and repair models equal access to the oracle so that we can better analyze the tradeoff between increased sampling overhead and accuracy gained. Chen et al. (2023b) instead aim to show how oracles and different repair strategies can improve the final accuracy compared to current, standard, non-repair-based approaches. Thus, our findings are not actually contradictory to those of Chen et al. (2023b).
>
> 2. Generally speaking, our work focuses on exploring the efficacy of self-repair in depth, while Chen et al. focus on comparing a broader range of different self-debugging strategies in a few different domains. Our studies thus complement each other.
>
> Concretely, our work focuses on what Chen et al. calls the “Unit Test + Explanation” style of feedback, in which the model is provided with external feedback from a unit test suite but then has to generate its own explanation of why the test failed. We study the effect of this textual explanation in great detail through our experiments in sections 4.2 and 4.3, and show that it is the limiting factor in this type of self-repair. We also offer a substantive discussion of the effect of the hyper-parameters, which amongst other things highlights the need to obtain sufficiently diverse initial samples if self-repair is to be successful.
>
> The scope of our study is also limited to what Chen et al. call “Text-to-Python Generation” tasks, while they also consider code translation and SQL query generation tasks. However, we do so in both an easier setting (HumanEval) and a significantly more challenging setting where baseline performance is lower (APPS), while Chen et al. focus on the relatively easy MBPP dataset. As we show in section 4.1 and Appendix B, the relationship between task difficulty and self-repair performance is quite subtle, an insight which we would not have been able to show without analyzing multiple datasets in detail.
>
> ---
>
> We once again thank Vfe4 for your helpful feedback on the paper. We hope that this discussion has clarified how our work relates to that of Chen et al. (2023b), and in particular the slightly different experimental settings as well as our emphasis on depth of understanding. If you have any more questions or thoughts, please do not hesitate to share them with us - we look forward to continuing this conversation throughout the discussion period!

---

> > ### Comment · Reviewer_Vfe4 · 2023-11-22
> > **Thanks for addressing my concerns.**
> >
> > Thank you for providing detailed response and updating the paper to address some of the concerns with additional experiments and revised texts. I think the new version provides more clarity on contributions of the paper.

---

> > > ### Author Response · Authors · 2023-11-22
> > >
> > > We are very pleased to hear that we were able to address some of your concerns, and that you felt it was appropriate to update your score to recommend acceptance. Thanks for helping us strengthen the paper! :)

---

### Author Response · Authors · 2023-11-20
**An updated version of the paper addresses feedback, strengthens the results further with additional experiments, and resolves a data processing issue causing incorrect results for Code Llama on HumanEval**

We thank all the reviewers for their encouraging comments, thought-provoking questions and detailed feedback! We are especially glad to hear that the reviewers found the paper “well-written” (all reviewers), “properly organized” (PNGf) and that we did a “great job in highlighting the key limitations of the analysis” (Vfe4). When it comes to the methodology, we are pleased to hear that our problem formulation was seen as “insightful” (LPE6) and that the systematic fashion in which we do the comparison was “appreciate[d]” (PNGf). Finally, given the empirical nature of this work we are especially grateful that the reviewers found the experiments “well done” (LPE6), and in particular that our focus on the feedback quality was deemed a “novel perspective” (PNGf) which could “enable more interesting future ideas” (Vfe4).

As we enter the second week of the discussion period, we are excited to share an updated version of the paper in which we have addressed the feedback from the reviewers and added several new experiments that strengthen some of our original findings as outlined below. We do want to point out that in the process of reviewing our scripts and running more experiments, we did find one small data processing error that affects one of the Code Llama results; this has now been corrected in the paper. The new results do not affect any of the conclusions of the paper or any of the major claims, but we nonetheless feel it is important for reviewers to be aware of this change.

Concretely, the changes made to the paper are:
- In response to feedback about the language sometimes being a bit too strong/general, as well as the discussion we’ve had with PNGf01 about how real-world software engineering differs from the tasks we’ve focused on here, we have replaced the second paragraph in the Limitations section with a discussion of how this work relates to more realistic Software Engineering tasks. We have also made minor revisions to the introduction to make it even more explicit to the reader early on that we are studying only self-contained Python programs in this work, as well as some minor wording changes throughout to clarify the scope of our claims. These changes are in **blue**.
- We have added the following extra experiments:
    - Code Llama baseline and self-repair on APPS
    - Code Llama + GPT-3.5 on APPS
    - Code Llama + GPT-4 on HumanEval
- The extra experiments we have added agree with and further strengthen our key findings:
    - Code Llama also does not appear to benefit from self-repair on APPS (much like GPT-3.5)
    - Code Llama does see performance improvements when given feedback from GPT-3.5
    - On HumanEval, boosting Code Llama with feedback from GPT-4 significantly improves performance compared to when doing so with GPT-3.5, further suggesting that the feedback alone can make a tremendous difference.
- The data processing error mentioned above was harming Code Llama’s reported self-repair performance on HumanEval. This error has now been fixed; we find that Code Llama does benefit from self-repair on this dataset, but that the feedback stage still remains a bottleneck, as illustrated by the significant improvements of Code Llama when paired with GPT-4.
- We have updated the Experiments section and appendices to include these results; all changes are in **green**, including updated figures.
- To make room for these additions, we have done some very minor word-smithing throughout the text.

We once again thank the reviewers for their extremely helpful comments and suggestions. We believe that these changes have further strengthened what the reviewers already appear to have found to be a strong paper, and hope that the reviewers also found the individual responses insightful.

**If the reviewers have found this discussion satisfactory, we now kindly ask them to consider updating their scores accordingly. If, on the other hand, they believe further discussion is needed, we eagerly await their thoughts and comments!** 🙂

---

### Meta-Review · Area_Chair_JWQV · 2023-12-14

**Metareview:**

> This paper investigates the ability of large language models (specifically, Code Llama, GPT-3.5, and GPT-4) to perform self-repair which refers to the model's capacity to identify and correct mistakes in its own generated code. The paper analyzes the effectiveness of self-repair in addressing code generation errors, specifically on problems sourced from HumanEval or APPS datasets. The findings suggest that the gains achieved through self-repair are often modest and vary significantly across different subsets of data. In some cases, self-repair does not result in noticeable improvements. The paper proposes that this limitation might be due to the model's ability to provide feedback on its own code, and stronger models might enhance self-repair effectiveness. Additionally, the paper explores the impact of providing the model with feedback from human participants, showing that this feedback significantly benefits the self-repair process, even for the advanced GPT-4 model. The paper offers a brief qualitative analysis to shed light on why this human feedback is valuable in improving code repair performance. Overall, the study provides insights into the challenges and potential enhancements of self-repair mechanisms in large language models for code generation tasks.

The limitations of the paper are mostly on the experimental side: not comparing to higher pass@ scores, nor finetuning LLMs for program repair (context). The authors had a productive discussion period with updates to the paper. Overall, the paper is clear and a worthy addition to the ICLR proceedings.

**Justification For Why Not Higher Score:**

The limitations of the paper are mostly on the experimental side: not comparing to higher pass@ scores, nor finetuning LLMs for program repair (context).

**Justification For Why Not Lower Score:**

N/A

---

### Decision · Program_Chairs · 2024-01-16

Accept (poster)